# H³DP: Triply-Hierarchical Diffusion Policy for Visuomotor Learning

**Yiyang Lu**[1*], **Yufeng Tian**[4*], **Zhecheng Yuan**[1,2,3*],
**Xianbang Wang**[1], **Pu Hua**[1,2,3], **Zhengrong Xue**[1,2,3], **Huazhe Xu**[1,2,3]

[1] Institute for Interdisciplinary Information Sciences, Tsinghua University,
[2] Shanghai Qi Zhi Institute, [3] Shanghai AI Lab, [4] Harbin Institute of Technology
luyy24@mails.tsinghua.edu.cn, huazhe_xu@mail.tsinghua.edu.cn

Figure 1: **H³DP** can not only achieve superior performance across 44 tasks on 5 simulation benchmarks, but also handle long-horizon challenging manipulation tasks in cluttered real-world scenarios.

## Abstract

Visuomotor policy learning has witnessed substantial progress in robotic manipulation, with recent approaches predominantly relying on generative models to model the action distribution. However, these methods often overlook the critical coupling between visual perception and action prediction. In this work, we introduce **Triply-Hierarchical Diffusion Policy** (**H³DP**), a novel visuomotor learning framework that explicitly incorporates hierarchical structures to strengthen the integration between visual features and action generation. H³DP contains **3** levels of hierarchy: (1) depth-aware input layering that organizes RGB-D observations based on depth information; (2) multi-scale visual representations that encode semantic features at varying levels of granularity; and (3) a hierarchically conditioned diffusion process that aligns the generation of coarse-to-fine actions with corresponding visual features. Extensive experiments demonstrate that H³DP yields a $+\mathbf{27.5}\%$ average relative improvement over baselines across **44** simulation tasks and achieves superior performance in **4** challenging bimanual real-world manipulation tasks. Project Page: https://h3-dp.github.io/.

---

*Equal Contribution

# 1 INTRODUCTION

Visuomotor policy learning has emerged as a prevailing paradigm in robotic manipulation [6; 70; 5; 65; 64]. Existing approaches have increasingly adopted powerful generative methods, such as diffusion and auto-regressive models, to model the action generation process [36; 60; 13; 46; 24]. However, these predominant methods have focused primarily on separately refining either the representation of perception or actions, often overlooking establishing a tight correspondence between perception and action. In contrast, human decision-making inherently involves hierarchical processing of information from perception to action [20; 3]. The visual cortex extracts features in a layered fashion and performs hierarchical inference based on visual motion perception, ultimately resulting in the generation of structured motor behaviors [25; 4]. Inspired by this, we argue that enabling learned visuomotor agents to emulate such hierarchical behavior patterns is also critical for enhancing their decision-making capabilities.

Prior works have primarily focused on hierarchically modeling the action generation process alone [51; 15], without explicitly incorporating hierarchical structure throughout the whole visuomotor policy pipeline. In this paper, we present $\mathbf{H^3DP}$, a novel visuomotor policy learning framework grounded in three levels of hierarchy: input, representation, and action generation. This design reflects the hierarchical processing mechanisms that humans use the visual cortex to perceive environmental stimuli to guide motor behavior.

At the input level, to better leverage the depth information in modern robotic benchmarks and datasets [21; 29; 45; 14], $H^3DP$ moves beyond prior 2D approaches that primarily rely on RGB or simple RGB-D concatenation, which has shown limited effectiveness in prior work [65; 72]. We introduce **depth-aware layering** strategy that partitions the RGB-D input into distinct layers based on depth cues. This approach not only enables the policy to explicitly distinguish between foreground and background, but also suppresses distractors and occlusions [39; 1], thereby enhancing the understanding and reasoning of spatial structure in the cluttered visual scenarios.

For visual representation, to address the limitations of flattening image features into a single vector, which can discard some spatial structures and semantic information [17; 42; 27], $H^3DP$ employs **multi-scale visual representation**, where different scales capture features at varying granularity levels, ranging from global context to fine visual details.

In the action generation stage, $H^3DP$ incorporates a key inductive bias inherent to the diffusion process: the tendency to progressively reconstruct features from low-frequency to high-frequency components [41; 9; 58], by **hierarchical action generation**. Specifically, coarse visual features guide initial denoising steps to shape the global structure (low-frequency components) of action, while fine-grained visual features inform the later steps to refine precise details (high-frequency components). This establishes a tighter coupling between action generation and visual encoding, enabling the policy to generate actions that are semantically grounded in multi-scale perceptual features.

We validate $H^3DP$ through extensive experiments on **44** simulation tasks across **5** diverse benchmarks, where it surpasses state-of-the-art methods by a relative average margin of $+\mathbf{27.5\%}$. Furthermore, in real-world evaluations, we deploy bimanual robotic systems to tackle four challenging tasks situated in cluttered environments, involving high disturbances and long-horizon objectives. $H^3DP$ achieves a $+\mathbf{72.4\%}$ relative performance improvement over baselines in real-world scenarios.

# 2 RELATED WORK

**Visual imitation learning.** Numerous studies have proposed efficient policy learning algorithms from different aspects [6; 70; 61]. As a representative approach, to endow the learned policy multi-modality ability, Diffusion Policy [6] incorporates the diffusion process to better represent the action distribution. Based on DP, methods like DP3 [65; 66] and 3D-Actor [22], designed for point cloud inputs, enhance the policy's scene understanding by refining the visual representation. Consistency Policy [36] and ManiCM [32] modify the inference process to achieve the inference acceleration. However, these approaches focus solely on enhancing either the action generation or the visual feature extraction, without explicitly modeling the relationship between them. To address this issue, we

propose a hierarchical framework that couples multi-scale visual representations with the diffusion process, enabling a more structured integration between visual features and action generation.

**Leveraging hierarchical information for policy learning.** In the computer vision community, numerous studies have leveraged hierarchical information to address a variety of downstream tasks [55; 59; 26; 40; 31; 43]. For example, standard diffusion models [49; 19; 48; 50] and flow matching [28; 30; 12] adopt the U-Net framework [42; 71], which exploits multi-scale feature representations to retain rich contextual information throughout the denoising process. VAR [53] innovatively employs multi-scale visual representations with quantization to perform image generation in an auto-regressive manner. In robot learning, recent works [15; 35; 69] have also begun to adopt hierarchical paradigms for policy learning. Dense Policy [51] leverages a bidirectional extension strategy to enable hierarchical action prediction. ARP [69] predicts a sequence of actions at different levels of abstraction in a hierarchical way. CARP [15] draws inspiration from VAR by employing a multi-scale VQ-VAE [55; 40] to construct action sequences and subsequently generating residual actions autoregressively using a GPT-style architecture [37]. However, these algorithms model only the hierarchical structure of the action generation process, without explicitly addressing the crucial linkage between visual representation and action in visuomotor policy learning. In contrast, H$^3$DP not only incorporates multi-scale visual representations but also leverages the inherent strengths of diffusion models to seamlessly integrate coarse-to-fine action generation into the diffusion process itself. Furthermore, by adopting a depth-aware layering strategy, H$^3$DP maximizes the utilization of hierarchical feature information across the input, latent, and output stages, thereby enriching the policy learning pipeline in a structured and semantically aligned manner.

## 3 METHOD

We employ three hierarchical structures to enhance the policy's understanding of visual input and predict more accurate action distributions as shown in Figure 3. At the input level, the RGB-D image is discretized into multiple layers to improve the policy's ability to distinguish and interpret foreground-background variations. Upon this, we adopt a multi-scale visual representation, wherein coarse-grained features capture global contextual information, while fine-grained features encode detailed scene attributes. On the action side, correspondingly, the representations at different scales are utilized to generate actions in a coarse-to-fine manner, thus strengthening the correlation between action and visual representations. Details of each component are presented in the following sections.

### 3.1 DEPTH-AWARE LAYERING

Effective robotic manipulation requires a strong grasp of 3D structure. RGB data provides texture and color, while depth encodes geometry such as object positions and distances. Combining them is powerful, but naive concatenation of RGB and depth often fails to improve performance [65; 72]. To better exploit depth, we partition the RGB-D image into $N$ non-overlapping layers based on depth values, as illustrated in Figure 2. Specifically, define $\{d_0 = d_{\min}, d_1, \ldots, d_N = d_{\max}\}$ as the depth boundaries for each layer. Image layer $I_m$ is formed by selecting pixels with depth in $[d_{m-1}, d_m)$, i.e.,

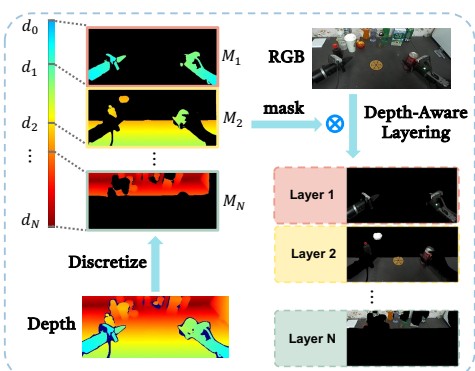

$$M_m^{(i,j)} = \mathbb{I}_{[d_{m-1} \leq D^{(i,j)} < d_m]}, \quad I_m = I \odot M_m, \quad (1)$$

where $I$ and $D$ are the RGB-D image and depth map, respectively, $\mathbb{I}$ is the indicator function. This representation separates the scene into meaningful foreground and background regions while preserving all visual details. It allows the policy to selectively attend to different depth planes, thereby improving both spatial perception and planning. We empha-

Figure 2: **Depth-aware layering.** We decompose the RGB-D image into $N$ non-overlapping layers based on depth, and encode each layer *independently* for better spatial understanding.

size its effectiveness in Section 4.3.2 and further compare against alternative discretization schemes in Section 4.4. Details for setting depth boundaries $\{d_m\}_{m=0}^N$ are provided in Appendix A.

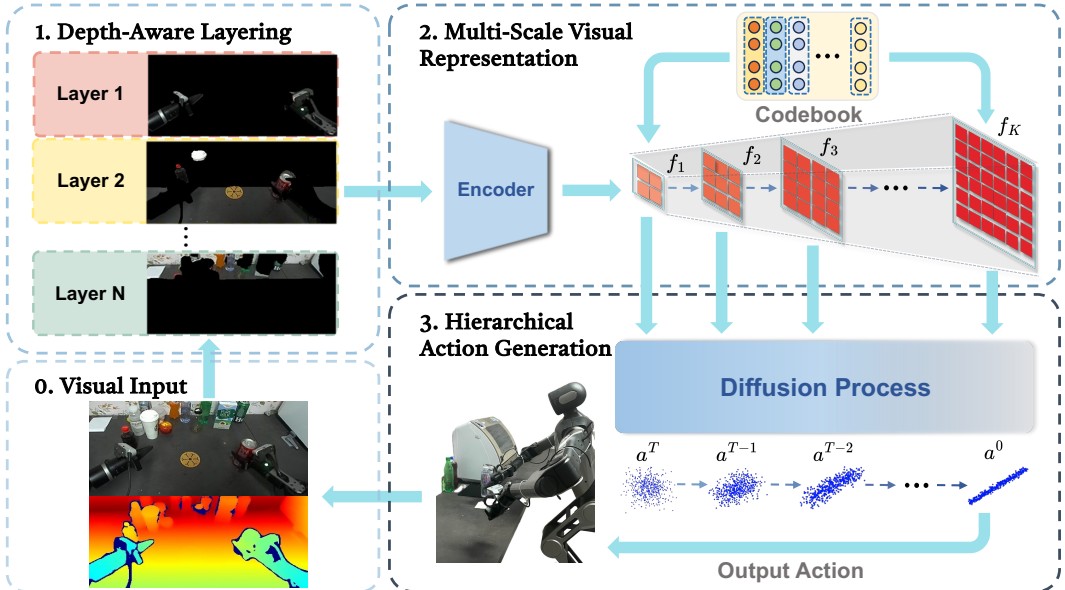

Figure 3: **Overview of H³DP**. H³DP integrates three hierarchical design principles across the perception and action generation pipeline. At the input level, RGB-D images are partitioned into multiple layers based on their depth values. Then, we employ multi-scale visual representations to capture features at varying levels of granularity. During the action generation, denoising process is divided into several stages guided by multi-scale visual representations.

## 3.2 MULTI-SCALE VISUAL REPRESENTATION

In visuomotor policy learning, visual representation plays a crucial role in embedding input images and mapping them to actions. An effective visual encoder should capture various granularity features of the visual scenarios and guide the policy to predict the action distribution. However, existing methods typically extract features at a single spatial scale or compress them into a fixed-resolution representation, limiting the expressiveness of learned features [17; 42; 27]. To address this problem, we hierarchically partition the feature map into multiple scales, enabling the capture of both coarse structural information and detailed fine-grained local cues.

**Interpolation and Quantization.** After applying depth-aware layering to the input image $I$, each layer $I_m$ is *independently encoded* into multi-scale feature maps $\{f_{m,k}|f_{m,k} \in \mathbb{R}^{h_k \times w_k \times C}\}_{k=1}^{K}$ via encoder $\mathcal{E}_m$, where $\{(h_k, w_k)\}_{k=1}^{K}$ denotes the spatial resolutions across scales. Adopting the quantization design in VQ-VAE [55; 40], these feature maps $\{f_{m,k}\}_{k=1}^{K}$ are quantized into discrete vectors drawn from a learnable codebook $\mathcal{Z}_m$. Specifically, each feature vector $f_{m,k}^{(i,j)}$ is mapped to its nearest neighbor in Euclidean distance, i.e., $f_{m,k}^{(i,j)} \leftarrow \underset{z \in \mathcal{Z}_m}{\arg\min} \|z - f_{m,k}^{(i,j)}\|_2$. By applying differentiable interpolation and lightweight convolution to the quantized features $f_{m,k}$, we then obtain the multi-scale visual representations $\{\hat{f}_{m,k}\}_{k=1}^{K}$ for each layer $I_m$. The pseudocode of full encoding procedure is detailed in Algorithm 1, Appendix B.

**Training.** To ensure consistent representations across scales, we aim to minimize the consistency loss between the original feature $f_m = \mathcal{E}_m(I_m)$ and the representation $\hat{f}_{m,k}$ at different scales:

$$\mathcal{L}_{\text{consistency}} = \sum_{m=1}^{N} \sum_{k=1}^{K} \left( \left\| \hat{f}_{m,k} - \text{sg}(f_m) \right\|_2^2 + \beta \left\| f_m - \text{sg}(\hat{f}_{m,k}) \right\|_2^2 \right), \tag{2}$$

where $\text{sg}(\cdot)$ is the stop gradient operator and $\beta$ balances the gradient flow between two terms. The visual encoder $\{\mathcal{E}_m\}_{m=1}^{N}$ and codebook $\{\mathcal{Z}_m\}_{m=1}^{N}$ are trained end-to-end, as described in detail in Appendix B. Notice that although the theoretical minmizer of the consistency loss leads to identical features across scales, in practice, due to the limited capacity of the codebook and downsampling operations, each scale captures distinct information. Coarse scales tend to retain global context, while finer scales preserve local details, serving as a strong inductive bias for the subsequent section.

## 3.3 HIERARCHICAL ACTION GENERATION

To match the inherent inductive biases of denoising process [41; 9; 58], we leverage multi-scale visual representations to model action generation in a coarse-to-fine manner. The early stage actions are derived from representations that capture global scene information, while fine-grained representations are responsible for generating detailed action components. This approach couples the visual representation and the action generation process via reinforcing their correspondence at the same hierarchical levels.

**Inference.** Our action generation module is a denoising diffusion model conditioned on multi-scale features $F = \{\hat{f}_k = \{\hat{f}_{m,k}\}_{m=1}^{N}\}_{k=1}^{K}$ and robot poses $q$. The denoising process unfolds over $T$ steps partitioned into $K$ stages $\cup_{k=1}^{K}(\tau_{k-1}, \tau_k]$. When $t \in (\tau_{k-1}, \tau_k]$, the denoising network $\epsilon_\theta^{(t)}$ conditioning on the corresponding feature map $\hat{f}_k$ and robot poses $q$, predicts the noise component $\epsilon^t = \epsilon_\theta^{(t)}(a^t|\hat{f}_k, q)$, then generates $a^{t-1}$ from $a^t$ via:

$$a^{t-1} = \alpha_t a^t + \beta_t \epsilon^t + \sigma_t \tilde{\epsilon}^t, \tag{3}$$

gradually transforming the Gaussian noise $a^T$ into the noise-free action $a^0$, where $\alpha_t, \beta_t, \sigma_t$ are fixed parameters depending on the noise scheduler, and $\tilde{\epsilon}^t \sim \mathcal{N}(0, \mathbf{I})$ is a Gaussian noise, see Appendix A for more details.

**Training.** To train the denoising network $\epsilon_\theta^{(t)}$, we randomly sample an observation-action pair $((I, q), a^0) \in \mathcal{D}$ and noise $\epsilon \sim \mathcal{N}(0, \mathbf{I})$. The network is optimized to predict $\epsilon$ given a noisy action $a^t = \sqrt{\gamma_t}a^0 + \sqrt{1 - \gamma_t}\epsilon$, via the objective:

$$\mathcal{L}_{\text{diffusion}} = \mathbb{E}_{a^0, \epsilon, t}\left[\|\epsilon_\theta^{(t)}(a^t|\hat{f}_K, q) - \epsilon\|^2\right]. \tag{4}$$

More implementation details can be found in Appendix A. By conditioning on the final feature $\hat{f}_K$ during training, gradients from the loss propagate through the entire hierarchical encoder, implicitly optimizing all $\{\hat{f}_k\}_{k=1}^{K}$. This design promotes consistency of representations at each scale for action generation while enhancing training efficiency.

**Discussions.** Diffusion models inherently aim to predict the posterior average of the target distribution conditioned on the provided features [8; 52], i.e., the optimal denoising network $\epsilon_{\theta*}^{(t)}$ follows $\epsilon_{\theta*}^{(t)}(a^t|f, q) = \mathbb{E}_{t, \epsilon, a^0, \sqrt{\gamma_t}a^0 + \sqrt{1-\gamma_t}\epsilon = a^t}[\epsilon|a^t, f, q]$. Earlier stages of the denoising process, characterized by higher noise levels, tend to have a posterior average with lower frequency, while later stages, with reduced noise, correspond to higher frequency components.

Our design of hierarchically conditioned diffusion aligns well with this property. Features at varying resolutions retain information across distinct frequency domains. Consequently, they provide robust guidance for generating specific frequency components of the action during relevant stages of the denoising process. Related experiments are shown in Section 4.1.3. By using lower-resolution features for earlier stages and gradually refining the predictions with higher-resolution features, the model benefits from both the stability of coarse representations and the precision of fine details.

## 4 EXPERIMENTS

In this section, we present extensive experiments across simulated and real-world settings to demonstrate the efficacy of H³DP. In addition, we perform thorough ablation analyses to evaluate the contribution of each hierarchical design, and further investigate the efficiency and effectiveness of our method in extracting visual representations.

### 4.1 SIMULATION EXPERIMENTS

#### 4.1.1 EXPERIMENT SETUP

**Simulation benchmarks and baselines:** To sufficiently verify the effectiveness of H³DP, we evaluate H³DP on **5** simulation benchmarks, encompassing a total of **44** tasks. These tasks span a variety of manipulation challenges, including articulated object manipulation [2; 38; 63], deformable object

Table 1: **Simulation task results.** Across 5 simulation benchmarks with various difficult levels, $H^3DP$ obtains $+\mathbf{27.5}\%$ relative performance gains on average over 44 tasks.

| Method \ Tasks | MetaWorld (Medium 11) | MetaWorld (Hard 5) | MetaWorld (Hard++ 5) | ManiSkill (Deformable 4) | ManiSkill (Rigid 4) | Adroit (3) | DexArt (4) | RoboTwin (8) | **Average** (**44**) |
|---|---|---|---|---|---|---|---|---|---|
| **$H^3DP$** | **98.3** | **87.8** | **95.8** | **59.3** | **65.3** | **87.3** | 53.3 | **57.4** | **$75.6_{\pm 18.6}$** |
| DP | 78.2 | 52.6 | 58.0 | 22.3 | 27.5 | 79.0 | 44.3 | 22.8 | $48.1_{\pm 23.1}$ |
| DP (w/ depth) | 77.7 | 57.2 | 71.2 | 44.5 | 40.8 | 76.0 | 42.0 | 12.6 | $52.8_{\pm 22.2}$ |
| DP3 | 89.1 | 52.6 | 88.4 | 26.5 | 33.5 | 84.0 | 54.8 | 45.9 | $59.3_{\pm 24.9}$ |

manipulation [16], bimanual manipulation [33], and dexterous manipulation [2; 38]. The details of the expert demonstrations can be found in Appendix C. To comprehensively assess the performance of $H^3DP$, we compare it against three baselines: *Diffusion Policy* [6], one of the most widely used visuomotor policy learning algorithms; *Diffusion Policy (w/ depth)*, which extends Diffusion Policy to incorporate RGB-D input to bridge the information gap; and *DP3* [65], an enhanced version of Diffusion Policy that leverages an efficient encoder for point cloud input.

**Evaluation metric:** Each experiment is conducted with 3 different seeds to mitigate performance variance. For each seed, we evaluate 20 episodes every 200 training epoches. In simpler MetaWorld, Adroit and DexArt tasks, we compute the average of the highest five success rates as its success rate, while in other environments, only the hightest success rate is recorded.

### 4.1.2 SIMULATION PERFORMANCE

As shown in Table 1, the simulation experiment results exhibit that $H^3DP$ outperforms or achieves comparable performance among the whole simulation benchmarks. Our method outperforms DP3 by a relative average margin of $+\mathbf{27.5}\%$. Notably, DP3 requires manual segmentation of the point cloud to remove background and task-irrelevant elements. This process introduces additional human effort and renders performance susceptible to segmentation quality. Relevant experimental results are provided in Appendix E.3.

In contrast, benefiting from our design, $H^3DP$ obtains superior performance using only raw RGB-D input, **without the need for any pre-processing or segmentation**. Furthermore, on the Adroit and DexArt benchmark, while DP3 leverages multi-view cameras to restore the complete point clouds, $H^3DP$ attains comparable performance using only one **single-camera** RGB-D image. The whole simulation results in each task can be found in Appendix E.1. Notably, the hierarchical design in $H^3DP$ introduces **negligible overhead** relative to DP3, while being significantly more efficient than DP as detailed in Appendix E.4.

### 4.1.3 SPECTRAL ANALYSIS OF ACTIONS

To gain a more comprehensive understanding of the action generation, we apply Discrete Fourier Transform (DFT) to examine how the frequency composition of actions evolves throughout the denoising process. Specifically, we conduct the analysis across 4 benchmarks and visualize the spectral characteristics of action chunks during generation. As shown in Figure 4, the results consistently indicate that the denoising process begins with the synthesis of low-frequency features, which are incrementally complemented by higher-frequency features in later stages. This observation not only shows that action, akin

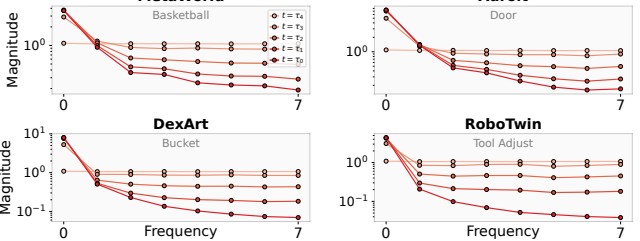

Figure 4: **Action DFT results.** As the denoising process progresses, the Gaussian noise ($t = \tau_4$) is gradually transformed into the predicted action ($t = \tau_0$). Timesteps $\tau_i$ is arranged in descending order of noise levels. The results reveal a consistent frequency evolution pattern: low-frequency components predominantly emerge during the early stages of denoising, whereas high-frequency features are progressively introduced in the latter phases of the process.

to image, exhibits an intrinsic inductive bias in the diffusion process, but also elucidates the action generation mechanism of $H^3DP$, wherein actions are hierarchically composed to captured features across varying levels of granularity.

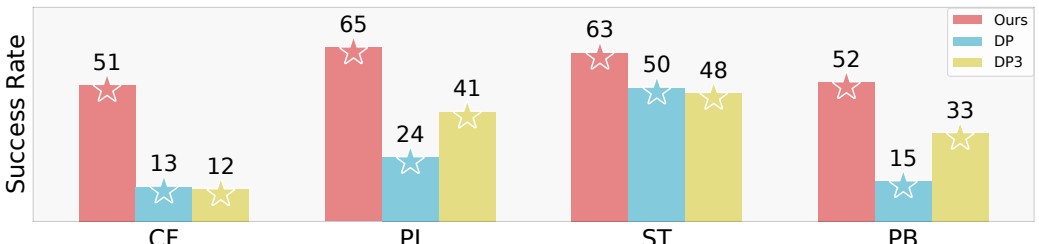

Figure 5: **Success rate in real-world.** We evaluate the success rate of H³DP, DP and DP3 across 4 challenging real-world tasks. H³DP outperforms DP and DP3 in all 4 tasks.

## 4.2 REAL-WORLD EXPERIMENTS

In terms of real-world experiments, we choose Galaxea R1 robot as our platform. We design four diverse challenging real-world tasks to evaluate the effectiveness of our method:

**Clean Fridge (CF):** In a cluttered refrigerator environment, the robot is required to relocate a transparent bottle from the upper compartment to the lower one. The bottle is randomized within a 30 cm × 5 cm region on both the upper and lower shelves of the refrigerator.

**Pour Juice (PJ):** This is a long-horizon task. The robot is required to place a cup in front of a water dispenser, scoop a spoonful of juice powder, then fill the cup with water, and finally put a straw in the cup. The cup is placed within a 7 cm × 7 cm area, and both the color of the juice powder and the position of the water dispenser are subject to variation across trials.

**Sweep Trash (ST):** This long-horizon task entails picking up a broom, sweeping scattered debris on a table into a dustpan, and subsequently emptying the contents into a trash bin. The trash is randomly distributed across the entire table surface, approximately within a 40 cm × 40 cm area.

**Place Bottle (PB):** The robot must place a bottle, initially located at a random position, onto a designated coaster. The bottle is placed within a 15 cm × 15 cm region, while the coaster is positioned within an around 25 cm × 25 cm area.

### 4.2.1 EXPERIMENT SETUP

We use the ZED camera to acquire the RGBD image. The demonstrations are collected by Meta Quest3. Each task is evaluated at 20 randomly sampled positions within the defined randomization range. We record the success trials and calculate the corresponding success rate. We compare H³DP with 2 baselines: *Diffusion Policy* [6] and *DP3* [65].

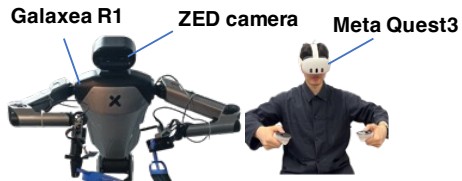

Figure 6: **Experiment Setup.**

In addition, during policy deployment, we adopt an *asynchronous* design to obtain an approximately double inference speed compared to baseline. We also introduce *temporal ensembling* and *p-masking* to improve temporal consistency and alleviate overfitting to the proprioception state. More details can be found in Appendix D.

### 4.2.2 EXPERIMENT RESULTS

**Spatial generalization:** As shown in Figure 5, H³DP significantly outperforms the baselines across all four real-world tasks, achieving an average relative improvement of +**72.4**%. It should be noted that in **CF** and **PJ** tasks, the policy is required to not only identify target objects in cluttered visual environments but also perform long-horizon reasoning to accomplish the tasks. While DP and DP3 struggle to complete either task, H³DP achieves substantial improvements. Therefore, H³DP demonstrates superior perceptual and decision-making capabilities compared to alternative algorithms. Furthermore, H³DP surpasses DP and DP3 even when trained with only **20**% of the expert data, as detailed in Appendix E.5. This finding highlights the efficiency of H³DP in learning from limited data.

Meanwhile, it should be noted that in terms of the point cloud based method DP3, it requires precise segmentation and high-fidelity depth sensing, resulting in it being less effective in handling our four cluttered real-world scenes that we designed.

Table 2: **Instance generalization results.** $H^3DP$ achieves $+21.0\%$ relative performance gain on average.

| Method \ Tasks | Place Bottle | | | Sweep Trash | | | Average |
| --- | --- | --- | --- | --- | --- | --- | --- |
| | coke bottle | sprite | can | $8\,cm^3$ | $64\,cm^3$ | $216\,cm^3$ | |
| **$H^3DP$** | **67** | **49** | **53** | **75** | **86** | 67 | 66.2 |
| DP | 25 | 16 | 28 | 52 | 72 | 60 | 42.2 |
| DP3 | 43 | 30 | 37 | 72 | 79 | **67** | 54.7 |

**Instance generalization:** Regarding instance generalization, we evaluate the model on **ST** and **PB** by varying the size and shape of bottles or trash items. As shown in Table 2, after replacing the objects with variants of differing sizes and shapes, $H^3DP$ maintains strong generalization capabilities attributable to its ability to hierarchically model features at multiple levels of granularity, and consistently outperforms baseline approaches across all settings.

## 4.3 ABLATION STUDY

In this section, we ablate each key component of our framework and conduct experiments on three benchmarks to further exhibit the effectiveness of $H^3DP$. Furthermore, to fully demonstrate the advantages of $H^3DP$, we also conduct additional experiments to analyze the efficiency and effectiveness of our method, especially in extracting task-relevant visual representations.

### 4.3.1 HIERARCHICAL DESIGNS

**Hierarchical design.** We ablate the three hierarchical components introduced in our framework and compare them against DP with RGB-D input. As shown in Table 3, each hierarchical component independently con-

Table 3: **Ablation on hierarchical features.**

| Methods \ Benchmarks | MW | MS | RT | Average |
| --- | --- | --- | --- | --- |
| **$H^3DP$** | **65.7** | **68.0** | **45.0** | **59.6** |
| w/o depth layering | 55.0 | 52.5 | 32.0 | 46.5 |
| w/o hierarchical action | 57.0 | 50.0 | 40.0 | 49.0 |
| w/o multi-scale representation | 53.7 | 52.5 | 40.0 | 48.7 |
| DP (w/ depth) | 46.7 | 47.5 | 32.0 | 42.1 |

tributes to performance improvement, consistently outperforming the DP (w/ depth). Furthermore, Table 3 also demonstrates that the integration of all three hierarchical designs leads to a substantial enhancement in overall performance.

**Impact of $N$ in depth-aware layering.** For the depth-aware layering component, we investigate whether the policy's performance is sensitive to the choice of the number of layers $N$. As presented in Table 4, our policy achieves optimal and comparable performance when $N$ is set to 3 or 4, a trend consistently observed across all evaluated benchmarks. When $N$ becomes excessively large, the image is over-

Table 4: **Ablation on number of layers $N$.**

| Methods \ Benchmarks | MW | MS | RT | Average |
| --- | --- | --- | --- | --- |
| $H^3DP$ ($N=1$) | 55.0 | 52.5 | 32.0 | 46.5 |
| $H^3DP$ ($N=2$) | 55.7 | 60.0 | 35.0 | 50.2 |
| $H^3DP$ ($N=3$) | 65.7 | **68.0** | 45.0 | **59.6** |
| $H^3DP$ ($N=4$) | **67.0** | 61.5 | **50.0** | **59.5** |
| $H^3DP$ ($N=5$) | 58.7 | 55.0 | 50.0 | 54.6 |
| $H^3DP$ ($N=6$) | 56.0 | 51.0 | 40.0 | 49.0 |

partitioned, thus reducing the representation capacity of the policy. Nevertheless, in such cases, the performance remains better than non-layered baseline. The results highlight the critical role of depth-aware layering in enhancing the policy's performance.

### 4.3.2 VISUAL REPRESENTATIONS

**Efficiency and Effectiveness of $H^3DP$ Encoder.** Prior work suggests that pre-trained visual representation may enhance spatial generalization of policy [61]. Hence, we investigate the impact of integrating a pre-trained visual encoder with the original DP. We specifically replace the standard ResNet encoder [18] in DP with DINOv2 [34] and evaluate on randomly selected tasks from the MetaWorld benchmark. The comparative results are presented in Table 5. Although DP-DINOv2 shows a marginal improvement on some tasks compared to the original DP baseline, this comes with longer training time, inference latency and larger number of parameters due to the DINOv2. In contrast, $H^3DP$ utilizes an efficient visual encoder with less than **0.7M** parameters, which achieves strong performance improvements over the original DP without incurring the aforementioned overheads. We show the effectiveness of adopting separate encoders for each depth layer in Appendix E.2.

Table 5: **Comparison with DP with pre-trained visual encoder.** While DP-DINOv2 yields small improvement after paying additional cost, H³DP demonstrates superior performance.

| Method \ Tasks | MetaWorld | | | | | | Average |
|---|---|---|---|---|---|---|---|
| | Hand Insert | Pick Out of Hole | Disassemble | Stick Pull | Soccer | Sweep Into | |
| **H³DP** | 100 | 40 | 96 | 83 | 85 | 100 | 84.0 |
| DP | 73 | 13 | 81 | 64 | 43 | 74 | 58.0 |
| DP-DINOv2 | 91 | 24 | 77 | 72 | 41 | 78 | 63.8 |

Table 6: **Performance comparison demonstrating the effectiveness of depth-aware layering.** Tasks with significant depth variations show great improvement only with depth layering compared to DP (w/ depth), surpassing the point cloud baseline (DP3).

| Method \ Tasks | MetaWorld | | | | | | Average |
|---|---|---|---|---|---|---|---|
| | Push | Shelf Place | Disassemble | Soccer | Pick Place Wall | Peg Insert Side | |
| **H³DP (only w/ depth layering)** | 100 | 95 | 98 | 55 | 100 | 86 | 89.0 |
| DP (w/ depth) | 79 | 29 | 76 | 37 | 80 | 53 | 59.0 |
| DP3 | 96 | 86 | 98 | 57 | 97 | 92 | 87.7 |

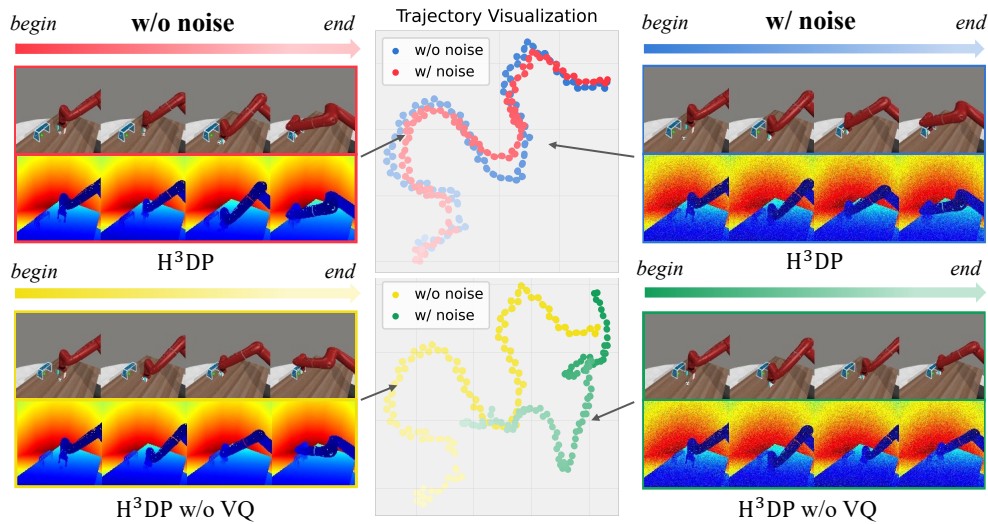

Figure 7: **Visualization of H³DP's robustness to noisy depth.** Under the noisy depth input, H³DP (blue) successfully finish the soccer while keeping the similar action trajectory as w/o noise (red). In contrast, H³DP w/o VQ (yellow) produces a deviated trajectory compared to the original (green).

**Effectiveness in Significant Depth-variant Tasks.** As introduced in Section 3.1, our depth-aware layering mechanism partitions the image into distinct layers. This layering offers a crucial advantage in scenarios with significant depth variations by providing a structured representation that preserves visual detail while emphasizing foreground-background separation. As seen in Table 6, our observations reveal a consistent pattern: in tasks involving significant depth variations, point cloud–based policy initially demonstrated superior performance compared to standard RGB-D processing, represented by DP (w/ depth). However, upon integrating the depth-aware layering mechanism, H³DP consistently outperforms the baseline on these tasks, which strongly supports our claim.

**Robustness to Noisy Depth.** To simulate low-quality depth input, we add Gaussian noise with a standard deviation of 0.1 to the normalized depth images (w/ noise in Table 7) during training and evaluation. As shown in Table 7, H³DP demonstrates strong robustness to degraded depth quality, maintaining high performance. In contrast, DP3 is highly sensitive to depth noise, exhibiting a significant drop in performance.

Table 7: **Ablation study on the effectiveness of VQ and robustness to noisy depth.** H³DP consistently outperforms DP3 especially under noisy depth input, demonstrating strong robustness to noisy depth input. VQ design further enhances the model's robustness to noisy depth input, leading to improved performance.

| Method \ Tasks | MetaWorld (w/o noise) | | MetaWorld (w/ noise) | |
|---|---|---|---|---|
| | Box Close | Soccer | Box Close | Soccer |
| **H³DP** | 98 | 85 | 98 | 75 |
| H³DP w/o VQ | 98 | 82 | 90 | 65 |
| DP3 | 78 | 57 | 28 | 17 |

Table 8: **Performance comparison with extensions of diffusion policy.** We compare H$^3$DP with flow-based, equivariance-based, and 3D representation-based approaches, respectively. H$^3$DP consistently outperforms these baselines across all evaluated tasks, highlighting its superior capability over various diffusion-based policy learning approaches.

| Method \ Tasks | MetaWorld | | | | | | Average |
|---|---|---|---|---|---|---|---|
| | Assembly | Pick Place | Shelf Place | Hand Insert | Pick Out of Hole | Push | |
| **H$^3$DP** | 100 | 99 | 100 | 100 | 40 | 100 | 89.8 |
| FlowPolicy [67] | 100 | 12 | 83 | 42 | 29 | 100 | 61.0 |
| ET-SEED [54] | 100 | 23 | 79 | 89 | 37 | 96 | 70.7 |
| 3D Diffuser Actor [22] | 100 | 0 | 65 | 33 | 26 | 100 | 54.0 |

Table 9: **Performance comparison with other policy with hierarchical structures**. We compare H$^3$DP with policies that utilize hierarchical designs. Our method outperforms these baselines across all evaluated tasks, demonstrating the effectiveness of our hierarchical designs.

| Method \ Tasks | MetaWorld | | | | | | Adroit | | DexArt | Average |
|---|---|---|---|---|---|---|---|---|---|---|
| | Bin Picking | Box Close | Hammer | Peg Insert Side | Disassemble | Shelf Place | Door | Pen | Laptop | |
| **H$^3$DP** | 100 | 98 | 100 | 98 | 96 | 100 | 79 | 83 | 81 | 92.8 |
| 2D Dense Policy | 25 | 51 | 86 | 60 | 71 | 59 | 59 | 65 | 28 | 56.0 |
| 3D Dense Policy | 47 | 69 | 100 | 82 | 98 | 77 | 72 | 61 | 85 | 76.7 |

To further understand the source of H$^3$DP's robustness to noisy depth, we compare it with a variant of H$^3$DP that does not use the feature vector quantization module (H$^3$DP w/o VQ). As shown in Table 7, this ablated version suffers from a substantial performance drop under the same noisy depth input. This indicates that the robustness of H$^3$DP can be largely attributed to the design of the feature vector quantization module. As shown in Figure 7, by mapping visual features to the nearest neighbor in a learned codebook, the codebook effectively projects representations back to the in-distribution space, thereby enhancing robustness to noise.

## 4.4 COMPARISON WITH MORE BASELINES

**Comparison with extensions of diffusion policy.** Previous discussions have primarily focused on comparing H$^3$DP with original diffusion formulation, i.e., DDPM-based DP [19]. Recently, flow-based [67; 7; 10], equivariance-based [57; 54; 62], and other 3D representation-based [65; 22] policy have been proposed to enhance the efficiency and effectiveness of diffusion-based policy learning. To provide a more comprehensive evaluation, we extend our comparisons to include additional state-of-the-art baselines. Specifically, we compare H$^3$DP with FlowPolicy [67] (flow-based), ET-SEED [54] (equivariance-based), and 3D Diffuser Actor [22] (3D representation-based) on selected tasks. As shown in Table 8, H$^3$DP consistently outperforms these baselines across all evaluated tasks, demonstrating its superior capability over various diffusion-based policy learning approaches.

**Comparison with other hierarchical methods.** To further validate the effectiveness of our hierarchical design, we compare H$^3$DP with Dense Policy [51], which predict action chunks coarse to fine via autoregressive modeling. As shown in Table 9, H$^3$DP outperforms Dense Policy by a relative average margin of $+\mathbf{21.0}\%$ across the evaluated tasks, demonstrating the superiority of our hierarchical designs over simple coarse-to-fine action modeling.

## 5 CONCLUSION

In this paper, we introduce H$^3$DP, an efficient generalizable visuomotor policy learning framework that can obtain superior performance in a wide range of simulations and challenging real-world tasks. Extensive empirical evidence suggests that establishing a more cohesive integration between visual representations and the action generation process can enhance the generalization capacity and learning efficiency of policies. The proposed three hierarchical designs not only facilitate the effective fusion of RGB and depth modalities, but also strengthen the correspondence between visual features and the generated actions at different granularity levels. In the future, we expect to extend the applicability of H$^3$DP to more intricate and fine-grained dexterous real-world tasks.

## 6 ETHICS STATEMENT

The proposed method in this paper is intended to enhance the capabilities of robotic systems in performing manipulation tasks. While the advancements in robotic manipulation can lead to significant benefits in various fields, including manufacturing, healthcare, and service industries, it is crucial to consider the ethical implications associated with the deployment of such technologies. Potential concerns include job displacement due to automation, privacy issues related to data collection, and the safety of human-robot interactions. It is imperative that researchers and practitioners in this field adhere to ethical guidelines and regulations to ensure that the development and application of robotic technologies are conducted responsibly and with consideration for societal impacts.

This paper has also benefited from the use of large language models (LLMs) to aid in refining and polishing the writing. LLMs were employed to enhance the clarity and coherence of the manuscript, ensuring that the ideas and contributions are communicated effectively. However, all technical content, experimental results, and conclusions were developed independently by the authors without reliance on LLMs.

## 7 REPRODUCIBILITY STATEMENT

To facilitate reproducibility, we will release the complete codebase, including the implementation of $H^3DP$, training scripts, and pre-trained models upon publication. The code will be well-documented to assist researchers in understanding and utilizing the framework effectively. Additionally, we will provide detailed instructions for setting up the environment and running experiments, along with the specific configurations used in our evaluations. All datasets and simulation environments referenced in this work are publicly available, and we will include links to these resources in the released code. Furthermore, we will share the hyperparameters and training protocols employed in our experiments to enable others to replicate our results accurately.

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

Table 10: **Hyperparameters used for MetaWorld, DexArt.**

| Hyperparameter | Value |
|---|---|
| Observation Horizon ($T_o$) | 2 |
| Action Horizon ($T_a$) | 2 |
| Prediction Action Horizon ($T_p$) | 4 |
| Optimizer | AdamW [23] |
| Betas ($\beta_1, \beta_2$) | [0.95, 0.999] |
| Learning Rate | 1.0e-4 |
| Weight Decay | 1.0e-6 |
| Learning Rate Scheduler | Cosine |
| Training Timesteps ($T$) | 50 |
| Inference Timesteps | 20 |
| Prediction Type | $\epsilon$-prediction |
| Image Resolution | $128 \times 128$ |
| Scale Number ($K$) | 4 |
| Multi-Scale Representation Resolutions ($\{(h_k, w_k)\}_{k=1}^K$) | $\{(1,1),(3,3),(5,5),(7,7)\}$ |
| Stage Boundaries ($\{\tau_k/T\}_{k=0}^K$) | $\{0,0.4,0.6,0.8,1.0\}$ |
| Codebook Size | 64 |

Table 11: **Hyperparameters used for Adroit.**

| Hyperparameter | Value |
|---|---|
| Observation Horizon ($T_o$) | 2 |
| Action Horizon ($T_a$) | 2 |
| Prediction Action Horizon ($T_p$) | 4 |
| Optimizer | AdamW |
| Betas ($\beta_1, \beta_2$) | [0.95, 0.999] |
| Learning Rate | 1.0e-4 |
| Weight Decay | 1.0e-6 |
| Learning Rate Scheduler | Cosine |
| Training Timesteps ($T$) | 50 |
| Inference Timesteps | 20 |
| Prediction Type | $\epsilon$-prediction |
| Image Resolution | $84 \times 84$ |
| Scale Number ($K$) | 4 |
| Multi-Scale Representation Resolutions ($\{(h_k, w_k)\}_{k=1}^K$) | $\{(1,1),(3,3),(5,5),(6,6)\}$ |
| Stage Boundaries ($\{\tau_k/T\}_{k=0}^K$) | $\{0,0.4,0.6,0.8,1.0\}$ |
| Codebook Size | 64 |

## APPENDIX

## A  HYPERPARAMETERS

To effectively address the varying levels of difficulty and distinct properties inherent to different benchmarks, we adapt our hyperparameter settings for each specific dataset. The chosen configurations, detailed in Table 10, 11, 12, 13, are selected based on previous works [6; 65; 72; 33].

In addition to the hyperparameters reported in the table, the choice of the number of layers $N$ demonstrates great importance, as shown in Table 4. Empirically, we choose $N = 4$ in Adroit, MetaWorld Hard and Hard++, and $N = 3$ in other benchmarks.

For the reverse process, we employ different formulations depending on the environment. The noise scheduler for diffusion process is determined by $\alpha_t$, defined using function $f(t)$

$$\alpha_t = \frac{f(t-1)}{f(t)}, \quad \text{where} \quad f(t) = \cos\left(\frac{\pi}{2}\frac{t}{T}\right), \tag{5}$$

Table 12: **Hyperparameters used for ManiSkill.**

| Hyperparameter | Value |
|---|---|
| Observation Horizon ($T_o$) | 2 |
| Action Horizon ($T_a$) | 8 |
| Prediction Action Horizon ($T_p$) | 16 |
| Optimizer | AdamW |
| Betas ($\beta_1, \beta_2$) | [0.9, 0.95] |
| Learning Rate | 1.0e-4 |
| Weight Decay | 1.0e-4 |
| Learning Rate Scheduler | One Cycle LR [47] |
| Training Timesteps ($T$) | 100 |
| Inference Timesteps | 100 |
| Prediction Type | $\epsilon$-prediction |
| Image Resolution | $128 \times 128$ |
| Scale Number ($K$) | 4 |
| Multi-Scale Representation Resolutions ($\{(h_k, w_k)\}_{k=1}^{K}$) | $\{(1,1),(3,3),(5,5),(7,7)\}$ |
| Stage Boundaries ($\{\tau_k\}_{k=0}^{K}/T$) | $\{0,0.4,0.6,0.8,1.0\}$ |
| Codebook Size | 64 |

Table 13: **Hyperparameters used for RoboTwin.**

| Hyperparameter | Value |
|---|---|
| Observation Horizon ($T_o$) | 3 |
| Action Horizon ($T_a$) | 2 |
| Prediction Action Horizon ($T_p$) | 8 |
| Optimizer | AdamW |
| Betas ($\beta_1, \beta_2$) | [0.95, 0.999] |
| Learning Rate | 1.0e-4 |
| Weight Decay | 1.0e-6 |
| Learning Rate Scheduler | Cosine |
| Training Timesteps ($T$) | 100 |
| Inference Timesteps | 100 |
| Prediction Type | $\epsilon$-prediction |
| Image Resolution | $180 \times 320$ |
| Scale Number ($K$) | 4 |
| Multi-Scale Representation Resolutions ($\{(h_k, w_k)\}_{k=1}^{K}$) | $\{(1,3),(3,5),(5,7),(5,9)\}$ |
| Stage Boundaries ($\{\tau_k\}_{k=0}^{K}/T$) | $\{0,0.4,0.6,0.8,1.0\}$ |
| Codebook Size | 64 |

where $T$ is the total number of diffusion timesteps. In MetaWorld, Adroit and DexArt, we follow the DDIM [48] approach, formulating the reverse process as an ODE, which corresponds to setting $\sigma_t = 0$ for all $t$. In ManiSkill and RoboTwin, we follow the design of DDPM [19] and formulate the reverse process as a Variance Preserving (VP) SDE [50]. Correspondingly,

$$\gamma_t = \prod_{i=1}^{t} \alpha_i^2, \quad \beta_t = \sqrt{1 - \gamma_{t-1} - \sigma_t^2} - \sqrt{1 - \gamma_t}\sqrt{\frac{\gamma_{t-1}}{\gamma_t}} \quad (6)$$

The boundary in Equation 1 is set following linear-increasing discretization [68], which is

$$d_i = \frac{i(i+1)}{N(N+1)}(d_{\max} - d_{\min}) + d_{\min}, \quad i = 0, \dots, N, \quad (7)$$

where $d_{\min}$ and $d_{\max}$ are the minimum and maximum depth values in the input depth image, respectively. This discretization strategy allocates finer depth resolution to closer objects, which are typically more relevant for manipulation tasks, while providing coarser resolution for distant objects.

## B    METHOD DETAILS

This section outlines the implementation details of our multi-scale encoding. The encoder $\mathcal{E}_m$ for each depth layer $m$ adopts the architecture from VQGAN [11], ensuring strong representational capacity while preserving spatial information. We use interpolate to denote a differentiable resizing operation (e.g. bilinear or nearest-neighbor interpolation), which is crucial for enabling gradient flow during training. The function $\mathcal{Q}$ represents the quantization process detailed in Section 3.2. Finally, after interpolating a feature map $f_{m,k}$ to the highest resolution, we apply a lightweight convolutional network $\phi_{m,k}$ designed to help restore fine details from the potentially lower-resolution source features.

The pseudocode for this process is outlined in Algorithm 1.

---

**Algorithm 1:** Multi-scale Encoding

1  **Inputs:**  raw image $I$
2  **Hyperparameters:**  depth layer number $N$, scale number $K$, resolutions $\{(h_k, w_k)\}_{k=1}^{K}$
3  Partition image $I$ into $N$ images $\{I_m\}_{m=1}^{N}$ according to Equation 1
4  **for** $m = 1, \ldots, N - 1$ **do**
5      $f_m \leftarrow \mathcal{E}_m(I_m) \in \mathbb{R}^{h_K \times w_K \times C}$
6      **for** $k = 1, \ldots, K$ **do**
7          $f_{m,k} \leftarrow \text{interpolate}(f_m, h_k, w_k) \in \mathbb{R}^{h_k \times w_k \times C}$
8          $f_{m,k} \leftarrow \mathcal{Q}(f_{m,k})$
9          $f_{m,k} \leftarrow \phi_{m,k}(\text{interpolate}(f_{m,k}, h_K, w_K)) \in \mathbb{R}^{h_K \times w_K \times C}$
10         $\hat{f}_{m,k} \leftarrow \sum_{k' \leq k} f_{m,k'}$
11         $f_m \leftarrow f_m - \hat{f}_{m,k}$
12 **Return:**  multi-scale features $F = \{\hat{f}_k = \{\hat{f}_{m,k}\}_{m=1}^{N-1}\}_{k=1}^{K}$

---

All trainable parameters, including the visual encoders $\{\mathcal{E}_m\}_{m=1}^{N-1}$, the codebooks $\{\mathcal{Z}_m\}_{m=1}^{N-1}$, the CNN parameters $\{\{\phi_{m,k}\}_{m=1}^{N-1}\}_{k=1}^{K}$, and the denoising network $\epsilon_\theta$, are trained jointly in an end-to-end manner. The optimization minimizes the combined objective function $\mathcal{L}$, defined as a weighted sum of consistency loss (Equation 2) and the diffusion loss (Equation 4):

$$\mathcal{L} = \mathcal{L}_{\text{diffusion}} + \alpha \mathcal{L}_{\text{consistency}}, \tag{8}$$

where $\alpha$ is a hyperparameter balancing the two loss terms.

## C    EXPERT DEMONSTRATIONS

Regarding the MetaWorld [63] and the RoboTwin [33] benchmarks, we utilize scripted policies to generate expert demonstrations. In the case of ManiSkill [16] tasks, we employ the officially provided demonstrations. Trajectories for other simulation benchmarks are collected with agents trained by RL algorithms [65; 44; 56]. The expert policies are evaluated over 200 episodes, and their success rates are detailed in Table 23.

Given the varying difficulty levels across benchmarks, we provide a different number of demonstrations for each. Specifically, we provide 50 trajectories per task for MetaWorld, Adroit, and RoboTwin. For DexArt, we follow the setup in [65] and provide 100 trajectories per task. For ManiSkill, we use all official demonstrations: 1000 for rigid tasks and 200 for deformable tasks.

In real-world experiments, we collect demonstrations of varying quantity, depending on the complexity and horizon length of the tasks. For short-horizon tasks, the number of collected trajectories is relatively limited — 100 for Clean Fridge and 200 for Place Bottle. In contrast, long-horizon tasks demand more comprehensive data coverage. We collect more demonstrations: 270 for Pour Juice and 500 for Sweep Trash. These demonstrations play a crucial role in guiding the training process, especially in scenarios where exploration is challenging or unsafe.

Table 14: **Comparison of real-world inference speeds for different methods.** The asynchronous version of our method demonstrates a significant speed-up by decoupling inference from action execution.

| Method | DP | DP3 | H$^3$DP | H$^3$DP (asynchronous) |
|---|---|---|---|---|
| Inference Speed (FPS) | 12.4 | 12.7 | 12.1 | 24.2 |

## D  REAL-WORLD TRAINING DETAILS

As mentioned in [15], DP-based methods often suffer from low inference speed, which can cause the inference process to stall. Prior approaches, including DP3 [65], attempt to address this by increasing action horizon (e.g. $T_a = 4$ or $T_a = 8$) or reducing the number of model parameters (e.g. Simple DP3). However, these strategies often compromise manipulation accuracy and dexterity. A further complication is that increasing $T_a$ widens the temporal gap between consecutive inference steps, leading to greater discrepancies in observed information, and consequently, divergence in predicted actions. This often results in noticeable jitter and discontinuities in manipulation.

In general, DP-based methods are hindered by low inference speed, temporal inconsistency and overfitting to proprioceptive information. To address these challenges and improve real-world performance, we employ several empirical techniques.

### D.1  HIGHER INFERENCE SPEED

To mitigate slow inference rooted in DP, we adopt an *asynchronous* design, achieving a final inference frequency of 10-15 Hz. Instead of waiting for the execution of all predicted actions before initiating the next inference cycle, our method performs inference concurrently with action execution. The predicted action is stored in a queue to be executed at a fixed inference speed (10-15 Hz in practice, 12 Hz as average).

The inference speeds achieved in real-world scenarios are presented in Table 14. H$^3$DP (asynchronous) demonstrates a superior inference speed compared to standard DP [6] and DP3 [65], as well as our synchronous H$^3$DP implementation. In addition to this speed advantage, H$^3$DP features a shorter action sequence length ($T_a = 2$), which contributes to more dexterous manipulation capabilities.

### D.2  TEMPORAL CONSISTENCY

Having adopted the *asynchronous* design, we have obtained action sequences with overlapping time intervals. To ensure temporal smoothness and reduce discontinuities, we incorporate *temporal ensembling mechanism* from ACT [70]. As in ACT, H$^3$DP performs a weighted average of actions with the same timestep across multiple overlapping sequences. This ensembling mitigates the gap between actions inferred from slightly different observations and effectively reduces jitter.

### D.3  ALLEVIATE OVERFITTING

Similar to other real-world robotic systems, H$^3$DP is susceptible to overfitting on proprioceptive inputs, often neglecting the RGB-D information. This is evidenced by that the model generates similar actions regardless of variations in object positions. We hypothesize that this occurs because the simple, low-parameter MLP used to encode proprioception is easier to optimize than the more complex CNN used for RGB-D input, leading to reliance on the former.

To mitigate this, we introduce a *p-masking* strategy during training. This mechanism stochastically masks all proprioceptive inputs with probability $p$, which decays linearly over the training process. Specifically, for training timestep $t$ in a total horizon $T$, $p(t) = 1 - t/T$. This schedule encourages the model to rely more on RGB-D features early in training, helping it avoid early-stage overfitting and develop stronger visual grounding.

Table 15: **Effectiveness of separate encoders for each depth layer.** Using separate encoders for each depth layer yields better performance compared to sharing a single encoder across layers with negligible increase in model size.

| Tasks \ Method | MetaWorld | | | Average |
| --- | --- | --- | --- | --- |
| | Box Close | Pick Place | Stick Pull | |
| H³DP | **98** | **99** | **83** | **93.3** |
| H³DP (shared encoder) | 92 | 89 | 76 | 85.7 |

Table 16: **Comparison of DP3 under different segmentation qualities.** We compare DP3 success rates on selected tasks when provided with different segmentation qualities, highlighting significant performance degradation.

| Method \ Tasks | MetaWorld | | | | | | Average |
| --- | --- | --- | --- | --- | --- | --- | --- |
| | Push | Shelf Place | Stick Pull | Soccer | Bin Picking | Pick Place Wall | |
| H³DP | 100 | 100 | 83 | 85 | 100 | 100 | 94.7 |
| DP3 | **96** | **86** | **61** | **57** | **100** | **97** | **82.8** |
| DP3 (w/o ideal segmentation) | 89 | 26 | 48 | 29 | 50 | 84 | 54.3 |

# E    ADDITIONAL EXPERIMENT RESULTS

## E.1    SIMULATION RESULTS FOR EACH TASK

We present the simulation results for each task in Table 22, which serves as a supplement to Table 1. For each experiment, we report the average success rate over three different random seeds. The final average result is obtained by averaging across all benchmarks.

We also provide the training progress of 4 algorithms on 12 various tasks across 3 different benchmarks in Figure 9. The selected tasks span a range of difficulties and are included without cherry picking to provide an unbiased view of each algorithm.

## E.2    EFFECTIVENESS OF SEPARATE ENCODERS

As mentioned in Section 3.2, we adopt separate encoders for each depth layer to allow specialization across spatial regions. While this may appear redundant, our whole encoder is still lightweight ($< 0.7$M parameters), and the overall model remains smaller than DP. We found that sharing a single encoder across layers led to performance degradation, as shown in Table 15. This suggests that separate encoders help capture the distinct characteristics of each depth layer more effectively, while the negligible increase in model size does not impact efficiency.

## E.3    IMPORTANCE OF SEGMENTATION IN DP3

As highlighted in Section 4.1.2, DP3 relies on manual segmentation of point cloud for optimal performance. To demonstrate this dependency, we evaluate DP3's performance under two distinct segmentation conditions using randomly selected tasks from the MetaWorld benchmark.

We compare the following two scenarios: *DP3 with ideal segmentation*, which utilizes clean segmented point clouds containing only the robot and task-relevant objects, as implemented in the original DP3 algorithm; *DP3 without ideal segmentation*, which utilizes point clouds that are intentionally processed to include desk surface upon which objects rest, while other background elements are still removed. This configuration simulates common real-world scenarios where simple or automated segmentation rules might fail to perfectly isolate the task-relevant objects.

As shown in Table 16, DP3's performance degrades substantially when operating on point clouds without ideal segmentation. This result confirms that DP3 is highly sensitive to the quality of the input point cloud segmentation.

Table 20: **Validation on Special Tasks.** Success rates of different dimmed light levels *dimmed lighting* (place bottle).

| Dimmed light level | 0% | 10% | 20% | 30% | 40% | 50% |
|---|---|---|---|---|---|---|
| Success rates | 52 | 55 | 50 | 40 | 45 | 40 |

In contrast, H$^3$DP operates directly on raw image without requiring such pre-processing, thereby avoiding such failure mode and the associated need for careful, potentially manual, segmentation tuning, especially common in real-world scenarios.

In our setup, the head-mounted camera is a ZED, which produces relatively low-quality visual inputs. This hinders the direct application of DP3 in our experimental setting. To ensure a fair comparison, we evaluate both H$^3$DP and DP3 on four real-world tasks with same visual inputs.

### E.4 INFERENCE SPEED AND MODEL SIZE

As shown in Table 17, we evaluate the inference speed of different methods within simulated environments. The results indicate that the primary bottleneck of the inference speed of H$^3$DP lies in the diffusion process itself, whereas the additional operations introduced for processing visual inputs and managing multi-scale representations incur only minimal computational overhead. A corresponding analysis of inference speed in real-world scenarios is available in Appendix D.1.

Table 17: **Comparison of inference speeds for DP, DP3 and H$^3$DP in simulation tasks.** The result indicates that additional operations introduced in H$^3$DP are lightweight compared to the diffusion process.

| Method | DP | DP3 | H$^3$DP |
|---|---|---|---|
| Inference Speed (FPS) | 11.1 | 12.2 | 12.0 |

As shown in Table 18, although H$^3$DP introduces multi-scale representation learning, it remains lightweight with negligible overhead in size compared to DP3, and is even smaller than the DP baseline.

Table 18: **Comparison of model sizes for DP, DP3 and H$^3$DP.** H$^3$DP introduces negligible overhead compared to DP3 while being smaller than DP.

| Method | DP | DP3 | H$^3$DP |
|---|---|---|---|
| Model Size (M) | 389 | 255 | 261 |

### E.5 RESULTS IN LOW-DATA REGIME

To further demonstrate the sample efficiency of our approach, we evaluate the performance of H$^3$DP under limited expert demonstrations in real-world settings. In particular, we train H$^3$DP using only 20% of the available training data. As shown in Table 19, H$^3$DP consistently achieves superior results even in such low-data scenarios, highlighting its strong sample efficiency.

Table 19: **H$^3$DP in low-data regime.** H$^3$DP demonstrates strong performance with only 20% expert data compared to DP and DP3.

| Method \ Tasks | CF | PJ | ST | PB | **Average** |
|---|---|---|---|---|---|
| H$^3$DP | 51 | 65 | 63 | 52 | 57.8 |
| H$^3$DP (w/ 20% expert data) | 37 | 44 | 58 | 33 | 43.0 |
| DP3 | 12 | 41 | 48 | 33 | 33.5 |
| DP | 13 | 24 | 50 | 15 | 22.5 |

### E.6 VALIDATION ON SPECIAL TASKS

To fully validate the effectiveness of H$^3$DP, we conduct additional experiments on two special tasks: *occlusion-heavy cluttered* task (tool adjust), where several objects are stacked together, to show the advantage of depth-aware layering in handling complex spatial arrangements; *dimmed lighting* task (place bottle), where the ambient light is significantly reduced during inference, to demonstrate the robustness of our hierarchical visual representation.

The results are summarized in Table 20 and Table 21. In the *occulsion-heavy cluttered* task, H³DP outperforms both DP and DP (w/ depth) by a significant margin, highlighting its ability to effectively utilize depth information in complex scenes. In the *dimmed lighting* task, H³DP maintains robust performance even as lighting conditions deteriorate, demonstrating the resilience of its hierarchical visual representation.

Table 21: **Validation on Special Tasks.** Success rates of different methods *occlusion-heavy cluttered* (tool adjust).

| Method | DP | DP (w/ depth) | H³DP |
|---|---|---|---|
| Success Rate | 0 | 32 | 45 |

### E.7 VISUALIZATION

We provide more visualization of our depth-aware layering results in Figure 8.

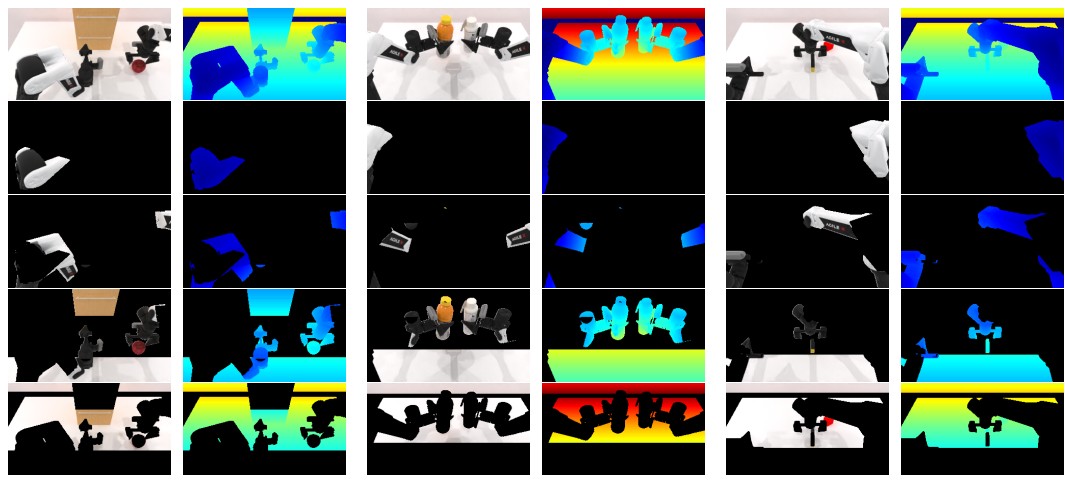

Apple Cabinet Storage      Dual Bottles Pick      Block Hammer Beat

Figure 8: Visualization of depth-aware layering on different tasks. From top to bottom: RGB image, depth image, and four layers obtained via depth-aware layering. Each layer captures different depth ranges, effectively segmenting the scene based on distance from the camera.

Table 22: **Success rates on 44 simulation tasks**. Results of four different methods for each task are provided in this table. The summary across domains is shown in Table 1.

| Method \ Tasks | MetaWorld [63] (Medium) | | | | | | | |
| --- | --- | --- | --- | --- | --- | --- | --- | --- |
| | Basketball | Bin Picking | Box Close | Coffee Pull | Coffee Push | Hammer | Soccer | Push Wall |
| **H³DP** | 100 | 100 | 98 | 100 | 100 | 100 | 85 | 100 |
| DP | 100 | 96 | 83 | 82 | 84 | 64 | 43 | 76 |
| DP (w/ depth) | 100 | 98 | 77 | 79 | 79 | 64 | 37 | 70 |
| DP3 | 100 | 100 | 78 | 100 | 100 | 97 | 57 | 95 |

| Method \ Tasks | MetaWorld (Medium) | | | MetaWorld (Hard) | | | | |
| --- | --- | --- | --- | --- | --- | --- | --- | --- |
| | Peg Insert Side | Sweep | Sweep Into | Assembly | Hand Insert | Pick Out of Hole | Pick Place | Push |
| **H³DP** | 98 | 100 | 100 | 100 | 100 | 40 | 99 | 100 |
| DP | 62 | 96 | 74 | 100 | 73 | 13 | 0 | 77 |
| DP (w/ depth) | 53 | 98 | 100 | 100 | 75 | 32 | 0 | 79 |
| DP3 | 92 | 100 | 61 | 100 | 37 | 30 | 0 | 96 |

| Method \ Tasks | MetaWorld (Hard++) | | | | | DexArt [2] | | | |
| --- | --- | --- | --- | --- | --- | --- | --- | --- | --- |
| | Shelf Place | Diassemble | Stick Pull | Stick Push | Pick Place Wall | Laptop | Faucet | Toilet | Bucket |
| **H³DP** | 100 | 96 | 83 | 100 | 100 | 81 | 34 | 70 | 28 |
| DP | 20 | 81 | 64 | 70 | 55 | 69 | 23 | 58 | 27 |
| DP (w/ depth) | 29 | 76 | 71 | 100 | 80 | 63 | 20 | 62 | 23 |
| DP3 | 86 | 98 | 61 | 100 | 97 | 80 | 33 | 79 | 27 |

| Method \ Tasks | Adroit [38] | | | ManiSkill [16] (Rigid) | | | |
| --- | --- | --- | --- | --- | --- | --- | --- |
| | Hammer | Door | Pen | Peg Insertion Side (Grasp) | Peg Insertion Side (Align) | Pick Cube | Turn Faucet |
| **H³DP** | 100 | 79 | 83 | 88 | 15 | 85 | 73 |
| DP | 95 | 69 | 73 | 78 | 7 | 17 | 8 |
| DP (w/ depth) | 100 | 66 | 62 | 93 | 12 | 33 | 23 |
| DP3 | 100 | 71 | 81 | 63 | 12 | 10 | 48 |

| Method \ Tasks | ManiSkill (Deformable) | | | | RoboTwin [33] | | |
| --- | --- | --- | --- | --- | --- | --- | --- |
| | Excavate | Hang | Pour | Fill | Apple Cabinet Storage | Dual Bottles Pick (Easy) | Dual Bottles Pick (Hard) |
| **H³DP** | 38 | 93 | 8 | 98 | 98 | 48 | 53 |
| DP | 2 | 52 | 0 | 36 | 73 | 53 | 28 |
| DP (w/ depth) | 23 | 78 | 7 | 72 | 2 | 33 | 25 |
| DP3 | 15 | 80 | 0 | 12 | 55 | 55 | 42 |

| Method \ Tasks | RoboTwin | | | | | Average |
| --- | --- | --- | --- | --- | --- | --- |
| | Block Handover | Block Hammer Beat | Diverse Bottles Pick | Pick Apple Messy | Tool Adjust | |
| **H³DP** | 70 | 85 | 25 | 35 | 45 | $75.6_{\pm 18.6}$ |
| DP | 28 | 0 | 0 | 0 | 0 | $48.1_{\pm 23.1}$ |
| DP (w/ depth) | 0 | 0 | 2 | 7 | 32 | $52.8_{\pm 22.2}$ |
| DP3 | 85 | 47 | 30 | 8 | 45 | $59.3_{\pm 24.9}$ |

Table 23: **Success rates of experts on 44 simulation tasks.** We evaluate 200 episodes for each task. For ManiSkill tasks, the demonstrations are provided officially, and we record the success rates as 100%. The final average result is obtained by averaging across all benchmarks.

| Method \ Tasks | MetaWorld [63] (Medium) | | | | | | | |
|---|---|---|---|---|---|---|---|---|
| | Basketball | Bin Picking | Box Close | Coffee Pull | Coffee Push | Hammer | Soccer | Push Wall |
| Expert | 100.0 | 97.0 | 90.0 | 100.0 | 100.0 | 100.0 | 90.5 | 100.0 |

| Method \ Tasks | MetaWorld (Medium) | | | MetaWorld (Hard) | | | | |
|---|---|---|---|---|---|---|---|---|
| | Peg Insert Side | Sweep | Sweep Into | Assembly | Hand Insert | Pick Out of Hole | Pick Place | Push |
| Expert | 92.0 | 100.0 | 90.0 | 100.0 | 100.0 | 100.0 | 100.0 | 100.0 |

| Method \ Tasks | MetaWorld (Hard++) | | | | | DexArt [2] | | | |
|---|---|---|---|---|---|---|---|---|---|
| | Shelf Place | Diassemble | Stick Pull | Stick Push | Pick Place Wall | Laptop | Faucet | Toilet | Bucket |
| Expert | 99.5 | 92.5 | 95.0 | 100.0 | 99.5 | 86.5 | 58.0 | 66.5 | 80.0 |

| Method \ Tasks | Adroit [38] | | | ManiSkill [16] (Rigid) | | | |
|---|---|---|---|---|---|---|---|
| | Hammer | Door | Pen | Peg Insertion Side (Grasp) | Peg Insertion Side (Align) | Pick Cube | Turn Faucet |
| Expert | 99.0 | 100.0 | 97.0 | 100.0 | 100.0 | 100.0 | 100.0 |

| Method \ Tasks | ManiSkill (Deformable) | | | | RoboTwin [33] | | |
|---|---|---|---|---|---|---|---|
| | Excavate | Hang | Pour | Fill | Apple Cabinet Storage | Dual Bottles Pick (Easy) | Dual Bottles Pick (Hard) |
| Expert | 100.0 | 100.0 | 100.0 | 100.0 | 96.0 | 97.0 | 55.5 |

| Method \ Tasks | RoboTwin | | | | | Average |
|---|---|---|---|---|---|---|
| | Block Handover | Block Hammer Beat | Diverse Bottles Pick | Pick Apple Messy | Tool Adjust | |
| Expert | 98.0 | 97.0 | 72.0 | 88.5 | 86.5 | 93.9 |

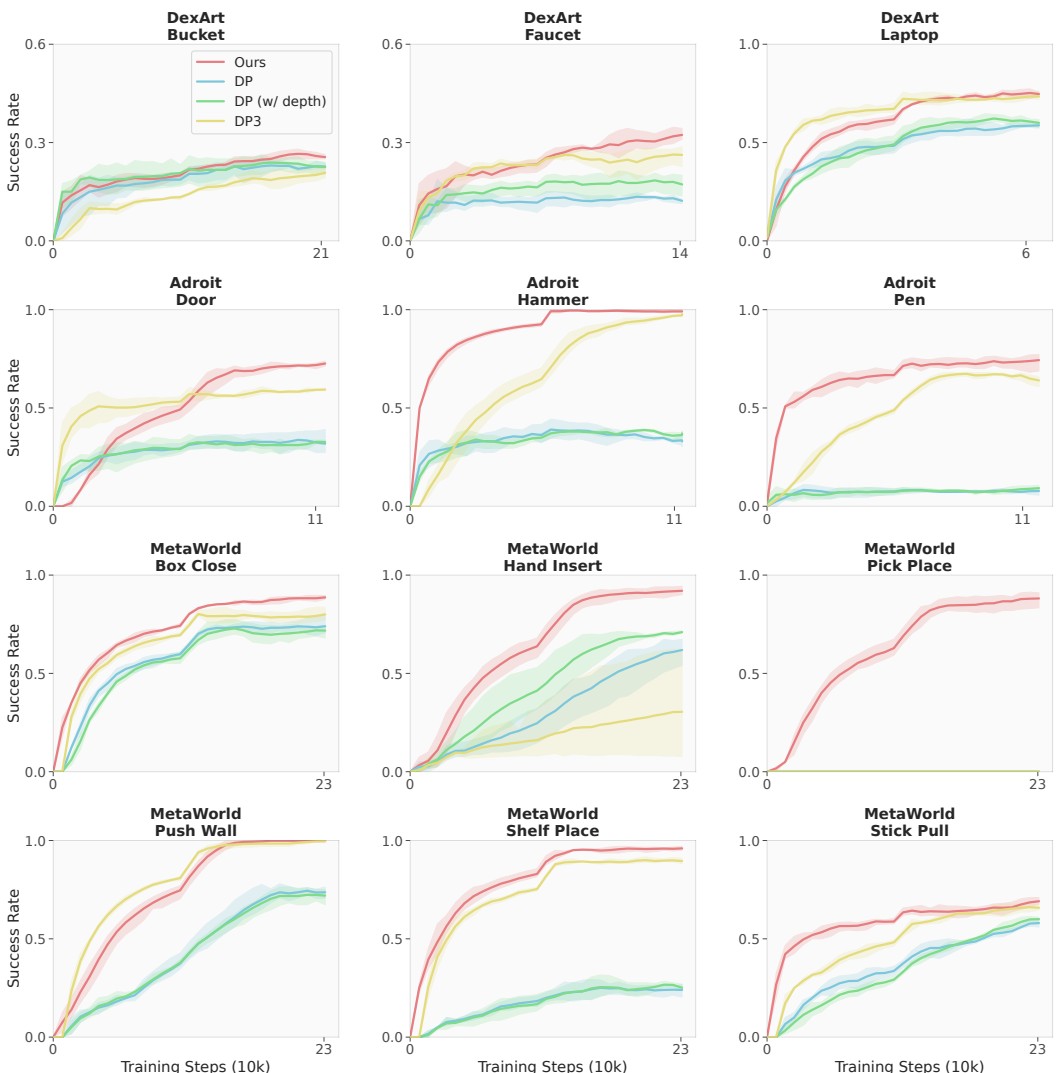

Figure 9: **Learning curves of the four methods on 12 randomly sampled diverse simulation tasks.** In most tasks, H³DP demonstrates faster convergence, higher final success rates, and lower variance compared to other three methods.

