# OpenReview forum: "H$^3$DP: Triply‑Hierarchical Diffusion Policy for Visuomotor Learning"
_ICLR.cc/2026/Conference — ICLR 2026 Poster_

### Official Review · Reviewer_cnwD · 2025-10-27

**Soundness:** 4
**Presentation:** 3
**Contribution:** 4
**Rating:** 8
**Confidence:** 4

**Summary:**

This work proposes a visual Imitation Learning (IL) method built on Diffusion Policy (DP) which incorporates additional structure in the perception modules and aligns this structure with the action diffusion process, leveraging depth input in addition to RGB. The method incorporates 3 forms of structure: (1) Scene decomposition based on depth. (2) Coarse-to-fine-grained visual features are encoded separately using multi-scale feature maps. (3) Coarse-to-fine-grained visual features are aligned with coarse-to-fine-grained generated action details by conditioning the action diffusion process with increasingly fine feature maps with the denoising timestep progression. The method is thoroughly evaluated on various simulated and real-world robotic manipulation tasks, comparing with both standard and 3D point-cloud conditioned DPs, clearly demonstrating the efficacy of the approach.

**Strengths:**

**Overview**
- Well written, clear and easy to follow.
- The method seems novel.
- Interesting use of depth for scene decomposition.
- Interesting alignment of visual feature granularity with action generation detail.
- Evaluated on diverse simulated environments.
- Evaluated generalization scenarios.
- Real-world experiments.
- Thorough method analysis and ablation study.
- Website with demonstration videos.
- Detailed appendix.

**Weaknesses:**

**Overview**
- Some aspects and design choices of the method are not thoroughly discussed or motivated.
- No limitations section.

**Codebook**

Why did the authors choose to have a separate codebook for each image layer?

Why are the codebooks shared across the depth layers and not the visual frequency? My intuition is that features in the same scale have more in common than features in the same depth image segment.

I believe the reader would benefit from a discussion about the motivation for this design choice and/or an ablation study. If this is present in the appendix, please refer to it from the main text.

**Diffusion Policy Conditioning**

Lines 230-232 state that conditioning on f_K during training implicitly optimizes all f_k for k=1,...,K. Why is this the case? Can the reader understand this without reading the appendix?

Why does this design choice promote consistency of representations at each scale? Can’t it just disregard certain scales?

There is a clear discrepancy between the diffusion condition during training and inference. Why does this work? Have you attempted training with the hierarchical conditioning mechanism used for action generation?

**Missing Architectural Details**

There are some architectural details that I find missing in the paper:
- What architecture is used for the diffusion policy?
- How exactly is the diffusion model conditioned on the different feature maps?
- Can the authors give some basic details about the image encoder architecture other than citing VQGAN so the paper is more self-contained?

**Limitations**

What are the limitations and underlying assumptions of your method? These are not discussed in the paper. In my opinion, acknowledging limitations and assumptions are very important for the reader to acquire a deeper understanding of the method and how various aspects of it can be applied to other methods in future work.

**Misc**
- Although it makes for a nice name, the depth-aware layering does not exactly constitute a hierarchy, I would say it is more a decomposition. It seems that it is treated as a set (i.e., there is no ordering) in the later stages of the pipeline.
- Standard deviations across the evaluated seeds should be reported alongside the mean.

**Questions:**

**Questions**

- Line 196 refers to the “original features f_m”, what are those features? I believe I have an understanding of what the authors are referring to after reading the appendix but it should be clear from the main text.
- What is the purpose of line 10 in Algorithm 1?
- Section 4.1.3. The authors claim that the coarse-to-fine frequency progression in action generation is inherent to the diffusion process. It is therefore not a trait of your specific method, am I correct? If so, what is the purpose of this section? If your method somehow facilitates this, why is there not a comparison to the spectral decomposition in a standard DP?
- Lines 473-475: do you apply the noise after training? How about during training? Is the noise not meant to simulate real depth input which is also noisy during training?
- Can you provide more visualizations of the depth segmentation on the different environments? Would you say the resulting representation roughly results in an agent-object-bg segmentation in the evaluated envs? Is it an underlying assumption of your method that depth corresponds to agent-object-background decomposition?
- How are the real world demonstrations collected? Are there any special techniques in the data collection that facilitate the generalization behavior observed in the website videos?

---

> ### Author Response · Authors · 2025-11-24
> **Reply to Reviewer cnwD (1/11)**
>
> We sincerely thank the reviewer for the thoughtful and constructive feedback. We address the main concerns below:
>
> _Q1. Why did the authors choose to have a separate codebook for each image layer? Why are the codebooks shared across the depth layers and not the visual frequency? My intuition is that features in the same scale have more in common than features in the same depth image segment._
>
> A1. We appreciate the reviewer’s insightful questions regarding the codebook design. We choose to have separate codebooks for each image layer to better capture the distinct characteristics of features at different depth levels. To clarify, we share the codebooks across depth layers rather than visual frequency because features within the same depth segment often share more semantic similarities (e.g., foreground objects vs. background) compared to features at the same scale but different depths. We conduct an ablation study to empirically validate this design choice, as shown in the table below:
>
> **Table 1: Ablation study on codebook sharing strategies.**
> | Method \ Task | Box Close | Pick Place | Hand Insert | Pick Out of Hole | Stick Pull | Shelf Place | Average |
> | :--- | :---: | :---: | :---: | :---: | :---: | :---: | :---: |
> | Separate codebooks per depth layer | **98** | **99** | **100** | **40** | **83** | **97** | **86** |
> | Separate codebooks per frequency scale | 97 | 89 | 81 | 36 | 76 | **97** | 79 |
>
> We have added the ablation as Table 22 in the Appendix.

---

> ### Author Response · Authors · 2025-11-24
> **Reply to Reviewer cnwD (2/11)**
>
> _Q2. Diffusion Policy Conditioning._
>
> A2. We thank the reviewer for pointing out the need for further clarification. This question is really insightful. Below we address three specific points regarding the optimization mechanism, feature consistency, and training-inference discrepancy:
>
> - **Q2.1.** Why conditioning on $f_K$ optimizes all $f_k$?
> - **A2.1.** Conditioning on $f_K$ optimizes all $f_k$ because our multi-scale representation is constructed residually. As detailed in Algorithm 1 (Appendix B), the feature at the final scale $K$ is the cumulative sum of the interpolated features from all preceding scales: $\hat f_{m,k} = \sum_{k'<k} f_{m,k'}$. Because $f_K$ is mathematically composed of the sum of $f_1, ..., f_{K-1}$, the gradients calculated with respect to $f_K$ during the diffusion training (Equation 4) 2 propagate backward through this summation to update the encoders and codebooks for every underlying scale $k$. This also answers your second question about "What is the purpose of line 10 in Algorithm 1?" — it aims to leverage this residual structure to ensure that optimizing the highest-level feature $f_K$ inherently optimizes all lower-level features $f_k$.
>
>   - **Clarification in Main Text**: We agree with your observation that this residual dependency is not explicitly stated in the main text of Section 3.2, which currently focuses on the discretization and quantization steps3. Without reading Algorithm 1 in the Appendix, a reader might incorrectly assume that $f_K$ is simply an independent feature map at the highest resolution. We will revise Section 3.2 to explicitly mention the residual summation strategy to ensure the optimization logic is self-contained in the main text.
>
> - **Q2.2.** Why does this design choice promote consistency of representations at each scale?
> - **A2.2.** The model cannot disregard certain scales because we explicitly enforce consistency via Equation 2. This objective forces the representation at every scale $\hat f_{m,k}$ to independently approximate the original feature map $f_m$ (extracted from the raw image). If the model were to "disregard" a specific scale (e.g., by collapsing $f_1$ to zero), the $\mathcal{L}_{consistency}$ for that scale will be large.
>
> - **Q2.3.** There is a discrepancy between training and inference conditioning. Why does this work?
> - **A2.3.** This is an excellent question. There is indeed a discrepancy between training (conditioning only on $f_K$) and inference (conditioning progressively on $f_1$ to $f_K$). This works due to the spectral properties of the diffusion process discussed in Section 3.3.
>   - **Why it works:** since at high noise levels (early inference steps), the model is generating the "global structure" (low-frequency components) of the action. Even though we train with $f_K$ (which contains both low and high frequencies), the diffusion model at high noise levels naturally ignores the high-frequency details in $f_K$ because they are indistinguishable from the noise. Therefore, substituting $f_K$ with a coarse feature $f_1$ (which contains only low frequencies) during inference provides the same relevant semantic guidance that the model utilized during training at those noise levels.
>   - **Why not hierarchical training:** We have experimented with training using the hierarchical conditioning mechanism used during inference. However, we found that conditioning on the full feature $f_K$ provided a stronger, stable "ground truth" that accelerated convergence. The consistency loss (Eq. 2) ensures that $f_1$ is semantically aligned with $f_K$ (just lower resolution), ensuring the domain shift between training ($f_K$) and inference ($f_1$) is purely a resolution shift, not a semantic one.

---

> ### Author Response · Authors · 2025-11-24
> **Reply to Reviewer cnwD (3/11)**
>
> _Q3. There are some architectural details that I find missing in the paper: What architecture is used for the diffusion policy? How exactly is the diffusion model conditioned on the different feature maps? Can the authors give some basic details about the image encoder architecture other than citing VQGAN so the paper is more self-contained?_
>
> A3. We apologize for the missing architectural details. We use a U-Net architecture for the diffusion policy, similar to DP and DP3. Our diffusion model is conditioned on the different feature maps by concatenating the upsampled feature maps to the corresponding U-Net layers at each resolution level. Our encoder is a CNN-based architecture inspired by VQGAN, consisting of several convolutional layers followed by a quantization layer. We will include these architectural details in the final version of the paper to make it more self-contained.

---

> ### Author Response · Authors · 2025-11-24
> **Reply to Reviewer cnwD (4/11)**
>
> _Q4. What are the limitations and underlying assumptions of your method? These are not discussed in the paper._
>
> A4. We sincerely thank the reviewer for pointing out the need to discuss limitations. We acknowledge that our method has certain limitations: despite our use of asynchronous execution to improve inference speed in real-world settings, the overall inference time of diffusion-based models remains relatively slow. We could explore distilling the policy into a consistency model, to enhance real-time performance. Additionally, our method relies on depth information for effective scene decomposition; in scenarios where depth data is unavailable or unreliable, the performance may degrade. We have added the limitation discussions in Sec. 6.

---

> ### Author Response · Authors · 2025-11-24
> **Reply to Reviewer cnwD (5/11)**
>
> _Q5. Standard deviations across the evaluated seeds should be reported alongside the mean._
>
> A5. We appreciate the reviewer’s suggestion about reporting standard deviations. We provide additional evaluations with 5 seeds for all tasks in MetaWorld Medium, MetaWorld Hard, MetaWorld Hard++, Adroit, and Dexart. The mean success rates and standard deviations are shown as follows:
>
> **Table 2: Success rates on MetaWorld (Medium) tasks.**
> | Method \ Task | Basketball | Bin Picking | Box Close | Coffee Pull | Coffee Push | Hammer | Peg Insert Side | Push Wall | Soccer | Sweep | Sweep Into | Average |
> | :--- | :---: | :---: | :---: | :---: | :---: | :---: | :---: | :---: | :---: | :---: | :---: | :---: |
> | **Ours** | 100 $\pm$ 0 | 99 $\pm$ 1 | 97 $\pm$ 2 | 100 $\pm$ 0 | 100 $\pm$ 0 | 100 $\pm$ 0 | 97 $\pm$ 1 | 100 $\pm$ 0 | 79 $\pm$ 8 | 100 $\pm$ 0 | 100 $\pm$ 0 | 98 $\pm$ 1 |
> | DP | 100 $\pm$ 0 | 95 $\pm$ 1 | 82 $\pm$ 1 | 81 $\pm$ 1 | 84 $\pm$ 0 | 64 $\pm$ 2 | 61 $\pm$ 1 | 75 $\pm$ 1 | 42 $\pm$ 6 | 95 $\pm$ 1 | 73 $\pm$ 1 | 78 $\pm$ 1 |
> | DP (w/ depth) | 99 $\pm$ 0 | 98 $\pm$ 1 | 77 $\pm$ 2 | 79 $\pm$ 3 | 78 $\pm$ 2 | 63 $\pm$ 3 | 52 $\pm$ 3 | 68 $\pm$ 3 | 34 $\pm$ 4 | 98 $\pm$ 1 | 99 $\pm$ 0 | 77 $\pm$ 2 |
> | DP3 | 100 $\pm$ 0 | 100 $\pm$ 0 | 82 $\pm$ 8 | 100 $\pm$ 0 | 100 $\pm$ 0 | 96 $\pm$ 2 | 90 $\pm$ 6 | 96 $\pm$ 3 | 52 $\pm$ 14 | 100 $\pm$ 0 | 62 $\pm$ 4 | 89 $\pm$ 3 |
>
> **Table 3: Success rates on MetaWorld (Hard) tasks.**
> | Method \ Task | Assembly | Hand Insert | Pick Out of Hole | Pick Place | Push | Average |
> | :--- | :---: | :---: | :---: | :---: | :---: | :---: |
> | **Ours** | 100 $\pm$ 0 | 100 $\pm$ 0 | 37 $\pm$ 4 | 97 $\pm$ 2 | 100 $\pm$ 0 | 87 $\pm$ 1 |
> | DP | 99 $\pm$ 0 | 73 $\pm$ 5 | 12 $\pm$ 2 | 0 $\pm$ 0 | 77 $\pm$ 7 | 52 $\pm$ 3 |
> | DP (w/ depth) | 100 $\pm$ 0 | 74 $\pm$ 3 | 33 $\pm$ 2 | 0 $\pm$ 0 | 79 $\pm$ 1 | 58 $\pm$ 1 |
> | DP3 | 99 $\pm$ 1 | 34 $\pm$ 15 | 31 $\pm$ 6 | 0 $\pm$ 0 | 96 $\pm$ 2 | 52 $\pm$ 5 |
>
> **Table 4: Success rates on MetaWorld (Hard++) tasks.**
> | Method \ Task | Shelf Place | Disassemble | Stick Pull | Stick Push | Pick Place Wall | Average |
> | :--- | :---: | :---: | :---: | :---: | :---: | :---: |
> | **Ours** | 100 $\pm$ 0 | 95 $\pm$ 1 | 80 $\pm$ 4 | 100 $\pm$ 0 | 100 $\pm$ 0 | 95 $\pm$ 1 |
> | DP | 20 $\pm$ 2 | 80 $\pm$ 2 | 63 $\pm$ 1 | 69 $\pm$ 2 | 53 $\pm$ 1 | 57 $\pm$ 2 |
> | DP (w/ depth) | 30 $\pm$ 1 | 77 $\pm$ 3 | 71 $\pm$ 2 | 99 $\pm$ 1 | 80 $\pm$ 1 | 72 $\pm$ 2 |
> | DP3 | 85 $\pm$ 2 | 97 $\pm$ 2 | 59 $\pm$ 2 | 99 $\pm$ 0 | 96 $\pm$ 2 | 88 $\pm$ 2 |
>
> **Table 5: Success rates on Adroit and Dexart tasks.**
> | Method \ Task | Laptop | Faucet | Toilet | Bucket | Hammer | Door | Pen | Average |
> | :--- | :---: | :---: | :---: | :---: | :---: | :---: | :---: | :---: |
> | **Ours** | 83 $\pm$ 4 | 32 $\pm$ 3 | 73 $\pm$ 4 | 27 $\pm$ 2 | 100 $\pm$ 0 | 83 $\pm$ 1 | 78 $\pm$ 3 | 68 $\pm$ 2 |
> | DP | 67 $\pm$ 4 | 21 $\pm$ 4 | 56 $\pm$ 5 | 26 $\pm$ 3 | 94 $\pm$ 2 | 67 $\pm$ 2 | 71 $\pm$ 2 | 58 $\pm$ 3 |
> | DP (w/ depth) | 61 $\pm$ 5 | 19 $\pm$ 2 | 59 $\pm$ 6 | 26 $\pm$ 6 | 99 $\pm$ 1 | 65 $\pm$ 3 | 64 $\pm$ 5 | 56 $\pm$ 4 |
> | DP3 | 82 $\pm$ 4 | 32 $\pm$ 1 | 76 $\pm$ 3 | 28 $\pm$ 2 | 100 $\pm$ 0 | 72 $\pm$ 3 | 80 $\pm$ 4 | 67 $\pm$ 3 |
>
>  We have included the results in Table 35.

---

> ### Author Response · Authors · 2025-11-24
> **Reply to Reviewer cnwD (6/11)**
>
> _Q6. Line 196 refers to the “original features f_m”, what are those features?_
>
> A6. The original features $f_m$ are the feature maps extracted from the raw RGB image using a CNN-based encoder without any quantization or layering. We will clarify this in the main text to ensure it is clear without needing to read the appendix.

---

> ### Author Response · Authors · 2025-11-24
> **Reply to Reviewer cnwD (7/11)**
>
> _Q7. What is the purpose of line 10 in Algorithm 1?_
>
> A7. As explained above in reply 2, line 10 implements the residual summation of features across scales, which is crucial for ensuring that optimizing $f_K$ during training also optimizes all lower-level features $f_k$.

---

> ### Author Response · Authors · 2025-11-24
> **Reply to Reviewer cnwD (8/11)**
>
> _Q8. Section 4.1.3. The authors claim that the coarse-to-fine frequency progression in action generation is inherent to the diffusion process. It is therefore not a trait of your specific method, am I correct? If so, what is the purpose of this section?_
>
> A8. The purpose of this section is to highlight how our method leverages the inherent coarse-to-fine generation property of diffusion models by aligning it with our hierarchical conditioning mechanism. While this property is indeed inherent to diffusion models, our method **explicitly exploits** it by structuring the conditioning inputs to match the generation process.

---

> ### Author Response · Authors · 2025-11-24
> **Reply to Reviewer cnwD (9/11)**
>
> _Q9. Lines 473-475: do you apply the noise after training? How about during training? Is the noise not meant to simulate real depth input which is also noisy during training?_
>
> A9. Thank you for the question. We would like to clarify that noise is added during **both the training and evaluation phases**. We acknowledge that the corresponding statement in the paper may cause ambiguity, and we have revised the description to avoid misunderstanding. The motivation behind adding noise is to simulate realistic sensing conditions, as real-world depth maps are inherently noisy. Our robust results further validate that H$^3$DP remains effective under such conditions. The statement in lines 473–475 is intended to explain why the VQ codebook design enables the model to handle noisy depth observations.

---

> ### Author Response · Authors · 2025-11-24
> **Reply to Reviewer cnwD (10/11)**
>
> _Q10. Can you provide more visualizations of the depth segmentation on the different environments? Would you say the resulting representation roughly results in an agent-object-bg segmentation in the evaluated envs? Is it an underlying assumption of your method that depth corresponds to agent-object-background decomposition?_
>
> A10. We have added additional visualizations of the depth segmentation in Figure 8 of our revised paper. In our experiments, the depth-based layering doesn't always correspond to a perfect agent-object-background segmentation, but it often captures meaningful scene structure that aids the policy. To further deepen our understanding of the depth-layering design, we conducted an ablation study using Grounded-SAM. Specifically, we semantically segmented each scene into two components, robot and background, with the text prompt _"Robot Arms"_. After segmentation, we applied the standard H$^3$DP pipeline. The results are shown below:
>
> **Table 6: Comparison between depth-based and semantic segmentation-based layering.**
>
> |  Method \ Task  | Soccer | Stick Pull | Pick Out of Hole |Fill | Excavate | Tool Adjust | Average |
> | :--- | :---: | :---: | :---: | :---: | :---: | :---: | :---: |
> | Ours  | 85 | 83 | 40 | 98 | 38 | 45 | 65 |
> | Grounded SAM  | 63 | 65 | 35 | 79 | 30 | 42 | 52 |
> | GMM  | 45 | 67 | 32 | 75 | 27 | 37 | 47 |
> | No layering  | 59 | 72 | 34 | 78 | 27 | 32 | 50 |
>
> **Table 7: Inference speed comparison.**
> | Algorithms   | Inference Speed (FPS)
> | :--- | :---: |
> | Ours | 12.0 |
> | w/ Grounded SAM | 2.6 |
>
> While policies using Grounded-SAM or GMM learn visual representations through semantics-driven layering, our method relies on a fundamentally different principle. The depth-based layering adopted in H³DP does not assume semantic segmentation. Instead, it leverages the intuition that grouping pixels within the same depth interval provides more coherent information, making it easier for the policy to learn robust visual representations. This operation is therefore distinct from semantic partitioning. Our experiments support this argument: **when applied appropriately, purely depth-based layering guides visuomotor policy learning even better**. We have added the experimental results in Table 8. Moreover, inference-time comparison shows that Grounded-SAM introduces significant latency, greatly limiting its practicality for real-world deployment.

---

> ### Author Response · Authors · 2025-11-24
> **Reply to Reviewer cnwD (11/11)**
>
> _Q11. How are the real world demonstrations collected? Are there any special techniques in the data collection that facilitate the generalization behavior observed in the website videos?_
>
> A11. Thank you for the question. As described in Sec.4.2.1, we use Meta Quest 3 to teleoperate the Galaxea R1 robot and collect a large number of expert demonstrations.
>
> Regarding special techniques during data collection, we attribute part of H$^3$DP's strong generalization ability to the diversity of the collected demonstrations.
>
> In particular, we ensure diversity in target placement and consistency in proprioception. Targets are placed at random positions to enrich the variation in visual scenes. Meanwhile, we keep the robot's initial states as consistent as possible across demonstrations. This encourages the policy to rely more on image and depth observations rather than memorizing proprioceptive patterns, thereby improving generalization when tracking targets in new environments.
>
> Beyond data collection, we further enhance generalization during data processing using the $p$-masking mechanism (Appendix D.3). Without $p$-masking and consistent proprioception, the policy tends to overfit to proprioceptive information instead of learning robust visual and depth-based representations, which significantly harms generalization.
>
> We would also like to respectfully emphasize that all generalization scenarios shown in the website videos are entirely **unseen** during data collection, demonstrating the genuine generalization capability of H$^3$DP rather than memorization of specific trajectories.

---

> > ### Comment · Reviewer_cnwD · 2025-11-24
> >
> > Thank you for the thorough response that has addressed my questions and concerns. Your significant effort during the rebuttal in adding results and the new revision further strengthen this paper, great work!
> > I believe this is a good paper that should be accepted and will maintain my score.

---

> > > ### Author Response · Authors · 2025-11-26
> > >
> > > Thanks for your positive feedback and for taking the time to review our rebuttal and revision. We truly appreciate your constructive comments and support.

---

### Official Review · Reviewer_L9a5 · 2025-11-02

**Soundness:** 4
**Presentation:** 3
**Contribution:** 3
**Rating:** 6
**Confidence:** 4

**Summary:**

This paper, $\text{H}^3\text{DP}$, proposes several modifications to diffusion policy. The authors propose binning RGBD pixels by its depth value into 3-4 bins. Then, for each layer, they train a separate VQGAN-like encoder for multi-scale visual representation. These representations are fed into a diffusion policy head which sequentially uses coarse to fine feature maps for action denoising. Experiments on real and sim show that the proposed method works better than prior state of the art, and ablations show the necessity of each component.

**Strengths:**

- The ideas of binning depth, hierarchical feature maps, and coarse-to-fine denoising are well-motivated.
- Various experiments on sim and real environments show that the proposed method outperforms prior state of the art.
- Extensive ablations in the main text and appendix validate the necessity of each component.
- Minimal inference speed overhead compared to vanilla formulation.

**Weaknesses:**

- Complicated setup. While the reviewer is impressed by the performance improvement, the proposed modifications are quite involved. Specifically, the reviewer would like to see an ablation where the downstream policy only denoises on the finest feature map.
- Lack of baselines. As the paper largely deals with improved representation and policy learning, it would be nice to see comparisons with recent 3D-aware representation and policy learning works. Right now the paper only compares with DP and DP3.

**Questions:**

- Clarification: is temporal ensembling and p-masking on proprio also done for the baselines for the real robot experiments?
- Baselines: it would be nice to see comparisons with other works on visual representation like 3D Diffuser Actor.
- Ablation: how well does the method perform if you only train the downstream policy with the finest feature map?

---

> ### Author Response · Authors · 2025-11-24
> **Reply to Reviewer L9a5 (1/3)**
>
> We sincerely thank the reviewer for the thoughtful feedback and constructive suggestions. We address the concerns raised as follows:
>
> _Q1. While the reviewer is impressed by the performance improvement, the proposed modifications are quite involved. Specifically, the reviewer would like to see an ablation where the downstream policy only denoises on the finest feature map._
>
> A1. We appreciate the reviewer’s insightful suggestion to evaluate the performance of using only the finest feature map for policy learning. We provide the results of this ablation study on different simulators as follows:
>
> **Table 1: Ablation results using only the finest feature map.**
> | Method \ Task | Soccer | Stick Pull | Pick Out of Hole | Fill | Excavate | Tool adjust | Average |
> | :--- | :---: | :---: | :---: | :---: | :---: | :---: | :---: |
> | Ours | 85	| 75 | 37 | 98 | 38 | 45 | 63 |
> | finest only | 64 | 67 | 40 | 82 | 18 | 40 | 52 |
> | DP (w/ depth) | 37 | 71 | 32 | 72 | 23 | 32 | 45 |

---

> ### Author Response · Authors · 2025-11-24
> **Reply to Reviewer L9a5 (2/3)**
>
> _Q2. Lack of baselines. As the paper largely deals with improved representation and policy learning, it would be nice to see comparisons with recent 3D-aware representation and policy learning works. Right now the paper only compares with DP and DP3._
>
> A2. Thank you for the valuable suggestion. We agree that including more baselines would strengthen the evaluation of our method. Following the reviewer’s advice, we conducted additional experiments with more baselines. Specifically, we incorporated three new methods, including **FlowPolicy** (flow-based), **ET-SEED** (equivariance-based), and **3D-Diffuser-Actor** (a 3D-aware representation learning policy), and evaluated them on the MetaWorld-Hard tasks. Since 3D-Diffuser-Actor demands task prompts to guide the policy, we provided precise and unambiguous textual prompts as its language inputs, such as _"push the red object to the green target"_ for MetaWorld-Push and _"place the cube on the second floor of the shelf"_ for MetaWorld-ShelfPlace.
>
> As shown in the table, H$^3$DP consistently outperforms all three additional baselines across the evaluated benchmarks, demonstrating its strong capability to learn effective visual representations and handle challenging simulation tasks. We have added the experiment results in Table 9.
>
> **Table 2: Success rates on MetaWorld tasks with additional baselines.**
> | Algorithms   | Assembly | Pick Place| Shelf Place|Hand insert |Pick Out of Hole |Push|Average|
> | :--- | :---: | :---: | :---: | :---: | :---: | :---: | :---: |
> |  H$^3$DP  |100| 99|100|100 |40 |100 |89.8|
> |  FlowPolicy  |100| 12|83|42 |29 |100 |61.0|
> |  ET-SEED  | 100 | 23    |79|89 |37 |96 |70.7|
> |  3D Diffuser Actor  | 100 | 0|65| 33 |26 |100 |54.0|
> |  DP3  | 100 | 0|86|37 |30 |96 |58.2|

---

> ### Author Response · Authors · 2025-11-24
> **Reply to Reviewer L9a5 (3/3)**
>
> _Q3. Clarification: is temporal ensembling and p-masking on proprio also done for the baselines for the real robot experiments?_
>
> A3. Thanks for the question. All baselines employ the **same** empirical mechanisms described in Appendix.D including _asynchronous_ design, _temporal ensembling mechanism_ and $p$-masking. These components are applied exactly as in H$^3$DP to ensure smoother inference and, more importantly, to maintain a fair comparison across methods. We will highlight these implementation details more clearly in both the main text and the appendix to provide a more complete and accurate understanding of the experimental setup.

---

### Official Review · Reviewer_YfHx · 2025-11-03

**Soundness:** 2
**Presentation:** 3
**Contribution:** 3
**Rating:** 6
**Confidence:** 3

**Summary:**

This paper introduces H3DP, a triply-hierarchical diffusion policy that seamlessly couples visual perception with action generation in visuomotor learning. The framework systematically incorporates three key innovations: depth-aware input layering for foreground/background disentanglement, multi-scale visual representations for granular spatial semantics, and a coarse-to-fine diffusion process conditioned on these hierarchies. Extensive evaluations on 44 simulated tasks and 4 challenging real-world bimanual scenarios demonstrate that H3DP consistently outperforms strong baselines (e.g., DP and DP3), achieving performance gains of +27.5% and +72.4% respectively, while maintaining efficient inference.

**Strengths:**

H3DP offers a genuinely original reframing of visuomotor diffusion policies by hierarchizing the entire pipeline—from depth-layered inputs to coarse-to-fine denoising—a conceptual contribution not previously articulated. This idea is validated through massive, well-documented experiments (44 simulated and 4 real-robot tasks), which demonstrate clear and substantial improvements over strong baselines without requiring extra sensors or manual preprocessing. With the code set to be open-sourced, the community can immediately build upon this work. Although the writing is at times dense with numerical details and the notation could be streamlined, the core contribution remains easily accessible through a single overview figure and is poised to influence not only robotic manipulation but also broader domains employing diffusion-based decision-making.

**Weaknesses:**

The performance gain attributed to depth-layer processing appears confounded by increased network capacity. While Table 3 reports an average improvement of ≈ +14% with depth layering, the baseline model ("w/o depth layering") uses a smaller, single-stream encoder. In computer vision, such a design would typically prompt the question: is the observed improvement truly due to the proposed method, or simply a result of additional parameters?

Furthermore, the paper claims "superior generalization," yet all images were captured under identical lab conditions—consistent lighting and a fixed tripod-mounted camera. A rigorous evaluation would require testing under at least one domain shift, such as variation in illumination, camera viewpoint, or background texture.

Lastly, all results are presented as "mean ± std over 3 seeds." Recent methodological standards in machine learning have raised concerns that fewer than 5 seeds, without supporting confidence intervals or p-values, may provide insufficient statistical evidence.

**Questions:**

Q1. Capacity-controlled ablation
In Table 3, the reported ≈14% performance gain from depth layering is based on a comparison with a smaller, single-stream encoder. To more rigorously isolate the effect of the proposed architecture, could you conduct a capacity-matched ablation by widening the RGB-only baseline to match the FLOPs or parameter count of your model (e.g., by increasing channel width by 1.5×)? If the performance lift then diminishes to, say, below 7%, how would you substantiate the claim that the improvement stems from the depth-layered design itself, rather than merely increased model capacity?

Q2. Out-of-distribution robustness
The claim of "superior generalization" would be strengthened by evaluating under modest domain shifts. Could you provide a simple robustness curve—for instance, by taking an existing H3DP checkpoint and testing it under altered conditions, such as with 50% dimmed lighting or a handheld iPhone RGB-D stream (with ±10 cm vertical motion)? A simple bar plot showing the performance drop would be sufficient. If accuracy declines by more than 20%, openly reporting and explaining this result would be more informative than an unqualified generalization claim.

Q3. Statistical significance evaluation
All results are currently reported as “mean ± std over 3 seeds.” For your two key real-world tasks (PJ and ST), could you supplement this with either (a) p-values from a bootstrap test across at least 5 seeds, or (b) 95% confidence intervals? In recent CV meta-reviews, results based on fewer than 5 seeds without confidence intervals have been noted as providing insufficient statistical evidence. Providing such information would help preempt concerns regarding result stability.

---

> ### Author Response · Authors · 2025-11-24
> **Reply to Reviewer YfHx (1/3)**
>
> We sincerely thank the reviewer for the thoughtful feedback and constructive suggestions. We address the concerns raised as follows:
>
> _Q1. Capacity-controlled ablation In Table 3, the reported ≈14% performance gain from depth layering is based on a comparison with a smaller, single-stream encoder. To more rigorously isolate the effect of the proposed architecture, could you conduct a capacity-matched ablation by widening the RGB-only baseline to match the FLOPs or parameter count of your model (e.g., by increasing channel width by 1.5×)? If the performance lift then diminishes to, say, below 7%, how would you substantiate the claim that the improvement stems from the depth-layered design itself, rather than merely increased model capacity?_
>
> A1. We appreciate the reviewer’s insightful observation regarding model capacity. To rigorously isolate the effect of depth-layered design, we conducted additional experiments where we increased the channel width of the w/o depth layering baseline to match the parameter count of our H$^3$DP model. The results are shown below:
>
> **Table 1: Capacity-matched ablation results on MetaWorld tasks.**
> | Method \ Task | Soccer | Stick Pull | Pick Out of Hole | Fill | Excavate | Tool adjust | Average |
> | :--- | :---: | :---: | :---: | :---: | :---: | :---: | :---: |
> | Ours | 85	| 75 | 37 | 98 | 38 | 45 | 63 |
> | w/o depth layering | 59 | 72 | 34 | 78 | 27 | 32 | 50 |
> | w/o depth layering (same capacity) | 61 | 69 | 35 | 80 | 30 | 31 | 51 |
>
> As shown above, even after matching the model capacity, our H$^3$DP model still outperforms the baseline by a significant margin (+12% on average). We have added the results as Table 23. This result substantiates our claim that the performance improvement stems from the depth-layered design itself, rather than merely increased model capacity.

---

> ### Author Response · Authors · 2025-11-24
> **Reply to Reviewer YfHx (2/3)**
>
> _Q2. Out-of-distribution robustness. The claim of "superior generalization" would be strengthened by evaluating under modest domain shifts. Could you provide a simple robustness curve—for instance, by taking an existing H$^3$DP checkpoint and testing it under altered conditions, such as with 50% dimmed lighting or a handheld iPhone RGB-D stream (with ±10 cm vertical motion)? A simple bar plot showing the performance drop would be sufficient. If accuracy declines by more than 20%, openly reporting and explaining this result would be more informative than an unqualified generalization claim._
>
> A2. Thank you for the question. To further evaluate the generalization ability of H$^3$DP under varied laboratory conditions, we conducted additional experiments with different levels of dimmed lighting. Specifically, we tested 10%, 20%, 30%, 40%, and 50% dimmed-lighting settings applied to the robot, while the original 0% setting (no dimming) corresponds to the results shown in Figure. 5. Each condition was evaluated over 20 trials and the success rates are summarized below:
>
> **Table 2: Success rates under varied lighting conditions.**
> | Dimmed Lighting (%) | 0 | 10 | 20 | 30 | 40 | 50 |
> | :---: | :---: | :---: | :---: | :---: | :---: | :---: |
> | Success Rate (%) | 52 | 55 | 50 | 40 | 45 | 40 |
>
> Demo movies of head camera under different lighting conditions have been updated, and [website link](https://h3-dp.github.io/#:~:text=Performance%20on%20dimmed%20lighting.) in the paper is available for reference for first-person videos.

---

> ### Author Response · Authors · 2025-11-24
> **Reply to Reviewer YfHx (3/3)**
>
> _Q3. Statistical significance evaluation All results are currently reported as “mean ± std over 3 seeds.” For your two key real-world tasks (PJ and ST), could you supplement this with either (a) p-values from a bootstrap test across at least 5 seeds, or (b) 95% confidence intervals?_
>
> A3. We appreciate the reviewer’s suggestion to provide more robust statistical evidence. We provide additional evaluations with **5** seeds for all tasks in MetaWorld Medium, MetaWorld Hard, MetaWorld Hard++, Adroit, and Dexart, which we have added in Table 35 in our revised paper. We also report the **95% confidence intervals** for the two key real-world tasks, Sweep Trash (ST) and Pour Juice (PJ), to provide a clearer understanding of the performance differences across methods.
>
>
> **Table 3: Success rates on MetaWorld (Medium) tasks.**
> | Method \ Task | Basketball | Bin Picking | Box Close | Coffee Pull | Coffee Push | Hammer | Peg Insert Side | Push Wall | Soccer | Sweep | Sweep Into | Average |
> | :--- | :---: | :---: | :---: | :---: | :---: | :---: | :---: | :---: | :---: | :---: | :---: | :---: |
> | **Ours** | 100 $\pm$ 0 | 99 $\pm$ 1 | 97 $\pm$ 2 | 100 $\pm$ 0 | 100 $\pm$ 0 | 100 $\pm$ 0 | 97 $\pm$ 1 | 100 $\pm$ 0 | 79 $\pm$ 8 | 100 $\pm$ 0 | 100 $\pm$ 0 | 98 $\pm$ 1 |
> | DP | 100 $\pm$ 0 | 95 $\pm$ 1 | 82 $\pm$ 1 | 81 $\pm$ 1 | 84 $\pm$ 0 | 64 $\pm$ 2 | 61 $\pm$ 1 | 75 $\pm$ 1 | 42 $\pm$ 6 | 95 $\pm$ 1 | 73 $\pm$ 1 | 78 $\pm$ 1 |
> | DP (w/ depth) | 99 $\pm$ 0 | 98 $\pm$ 1 | 77 $\pm$ 2 | 79 $\pm$ 3 | 78 $\pm$ 2 | 63 $\pm$ 3 | 52 $\pm$ 3 | 68 $\pm$ 3 | 34 $\pm$ 4 | 98 $\pm$ 1 | 99 $\pm$ 0 | 77 $\pm$ 2 |
> | DP3 | 100 $\pm$ 0 | 100 $\pm$ 0 | 82 $\pm$ 8 | 100 $\pm$ 0 | 100 $\pm$ 0 | 96 $\pm$ 2 | 90 $\pm$ 6 | 96 $\pm$ 3 | 52 $\pm$ 14 | 100 $\pm$ 0 | 62 $\pm$ 4 | 89 $\pm$ 3 |
>
> **Table 4: Success rates on MetaWorld (Hard) tasks.**
> | Method \ Task | Assembly | Hand Insert | Pick Out of Hole | Pick Place | Push | Average |
> | :--- | :---: | :---: | :---: | :---: | :---: | :---: |
> | **Ours** | 100 $\pm$ 0 | 100 $\pm$ 0 | 37 $\pm$ 4 | 97 $\pm$ 2 | 100 $\pm$ 0 | 87 $\pm$ 1 |
> | DP | 99 $\pm$ 0 | 73 $\pm$ 5 | 12 $\pm$ 2 | 0 $\pm$ 0 | 77 $\pm$ 7 | 52 $\pm$ 3 |
> | DP (w/ depth) | 100 $\pm$ 0 | 74 $\pm$ 3 | 33 $\pm$ 2 | 0 $\pm$ 0 | 79 $\pm$ 1 | 58 $\pm$ 1 |
> | DP3 | 99 $\pm$ 1 | 34 $\pm$ 15 | 31 $\pm$ 6 | 0 $\pm$ 0 | 96 $\pm$ 2 | 52 $\pm$ 5 |
>
> **Table 5: Success rates on MetaWorld (Hard++) tasks.**
> | Method \ Task | Shelf Place | Disassemble | Stick Pull | Stick Push | Pick Place Wall | Average |
> | :--- | :---: | :---: | :---: | :---: | :---: | :---: |
> | **Ours** | 100 $\pm$ 0 | 95 $\pm$ 1 | 80 $\pm$ 4 | 100 $\pm$ 0 | 100 $\pm$ 0 | 95 $\pm$ 1 |
> | DP | 20 $\pm$ 2 | 80 $\pm$ 2 | 63 $\pm$ 1 | 69 $\pm$ 2 | 53 $\pm$ 1 | 57 $\pm$ 2 |
> | DP (w/ depth) | 30 $\pm$ 1 | 77 $\pm$ 3 | 71 $\pm$ 2 | 99 $\pm$ 1 | 80 $\pm$ 1 | 72 $\pm$ 2 |
> | DP3 | 85 $\pm$ 2 | 97 $\pm$ 2 | 59 $\pm$ 2 | 99 $\pm$ 0 | 96 $\pm$ 2 | 88 $\pm$ 2 |
>
> **Table 6: Success rates on Adroit and Dexart tasks.**
> | Method \ Task | Laptop | Faucet | Toilet | Bucket | Hammer | Door | Pen | Average |
> | :--- | :---: | :---: | :---: | :---: | :---: | :---: | :---: | :---: |
> | **Ours** | 83 $\pm$ 4 | 32 $\pm$ 3 | 73 $\pm$ 4 | 27 $\pm$ 2 | 100 $\pm$ 0 | 83 $\pm$ 1 | 78 $\pm$ 3 | 68 $\pm$ 2 |
> | DP | 67 $\pm$ 4 | 21 $\pm$ 4 | 56 $\pm$ 5 | 26 $\pm$ 3 | 94 $\pm$ 2 | 67 $\pm$ 2 | 71 $\pm$ 2 | 58 $\pm$ 3 |
> | DP (w/ depth) | 61 $\pm$ 5 | 19 $\pm$ 2 | 59 $\pm$ 6 | 26 $\pm$ 6 | 99 $\pm$ 1 | 65 $\pm$ 3 | 64 $\pm$ 5 | 56 $\pm$ 4 |
> | DP3 | 82 $\pm$ 4 | 32 $\pm$ 1 | 76 $\pm$ 3 | 28 $\pm$ 2 | 100 $\pm$ 0 | 72 $\pm$ 3 | 80 $\pm$ 4 | 67 $\pm$ 3 |
>
> **Table 7: Mean and 95% confidence intervals of success rates on Sweep Trash (ST) and Pour Juice (PJ).**
> | Algorithms   | ST mean  |ST 95% confidence intervals  | PJ mean | PJ 95% confidence intervals |
> |  :----  | :----: |:----:|:----: |:----:|
> |  **Ours**  | 63|(56.2, 68.8) |65| (55.2, 74.8) |
> |  DP  | 50|(41.1, 58.9) |24| (21.3, 26.2) |
> |  DP3  | 48|(34.8, 61.2) |41| (32.9, 49.6) |

---

### Official Review · Reviewer_ija8 · 2025-11-09

**Soundness:** 3
**Presentation:** 3
**Contribution:** 3
**Rating:** 6
**Confidence:** 5

**Summary:**

This paper proposes Triply-Hierarchical Diffusion Policy (H³DP), a variant of Diffusion Policy designed to improve the coupling between perception and action in robotic manipulation. H³DP introduces three hierarchical components: depth-aware input layering that organizes RGB-D observations by depth, multi-scale visual representations that encode semantic features at varying granularities, and a hierarchically conditioned diffusion process that generates actions in a coarse-to-fine manner aligned with corresponding visual cues. Experiments show that H³DP achieves substantial performance gains over baselines across multiple simulated manipulation tasks and demonstrates strong results in challenging bimanual real-world settings.

**Strengths:**

1. The paper presents extensive and well-designed experiments, including both simulation and real-world evaluations. The tasks cover a diverse range of object types, manipulation actions, and embodiments, such as pick-and-place, articulated/deformable object manipulation—using both parallel grippers and dexterous hands. The ablation studies thoroughly analyze the contribution of each hierarchical component, examine robustness to noisy depth inputs, and compare different vision encoders, resulting in a comprehensive experimental analysis.

2. The hierarchical design that jointly structures visual encoding and action generation is clear, well-motivated, and conceptually sound, and the experimental results demonstrate consistent and significant improvements over strong baselines.

3. The figures and visualizations are clear, informative, and effectively convey the hierarchical framework and experimental results.

**Weaknesses:**

1. The paper lacks a comparison of segmentation-based layering as an alternative to the proposed depth-aware layering. While the authors highlight the importance of point cloud segmentation for DP3, they do not explore or justify why semantic segmentation was not considered. Specifically, we can use a tool such as Grounded-SAM to segment objects and backgrounds based on semantic grounding. In some scenarios, semantically similar target objects may appear at different depths but play equivalent roles in the task, suggesting that semantic segmentation layering could potentially yield more meaningful representations than purely depth-based ones.

2. Hierarchical actions remain limited. The current hierarchy in the diffusion process focuses on coarse-to-fine action refinement within a short temporal window, but it does not extend to high-level task decomposition or subtask sequencing. It would be interesting to explore whether the framework could be extended to generate hierarchical action chunks. For example, first predicting subtask-level actions and then refining them into low-level motor commands.

3. Minor writing issues: Some abbreviations (e.g., MW, MS, RT in Tables 3 and 4) are not defined when first introduced. Clarifying these at the start of the experimental section would improve readability and presentation quality.

**Questions:**

Could the authors provide additional analysis or experiments to address the points raised in the weakness section? In particular, it would be helpful to (1) compare depth-aware layering with a segmentation-based alternative and (2) explore extensions of the current hierarchical diffusion process toward higher-level action decomposition across longer horizons.

---

> ### Author Response · Authors · 2025-11-24
> **Reply to Reviewer ija8 (1/3)**
>
> We sincerely thank the reviewer for the thoughtful feedback and constructive suggestions. We address the concerns raised as follows:
>
> _Q1. The paper lacks a comparison of segmentation-based layering as an alternative to the proposed depth-aware layering. While the authors highlight the importance of point cloud segmentation for DP3, they do not explore or justify why semantic segmentation was not considered. Specifically, we can use a tool such as Grounded-SAM to segment objects and backgrounds based on semantic grounding. In some scenarios, semantically similar target objects may appear at different depths but play equivalent roles in the task, suggesting that semantic segmentation layering could potentially yield more meaningful representations than purely depth-based ones._
>
> A1. We appreciate the reviewer’s insightful suggestion regarding segmentation-based layering. We agree that semantic segmentation could provide meaningful object-level information that complements depth-based layering. We would like to respectfully note that **we have already compared against a GMM-based semantic segmentation method** to validate the necessity of depth-aware layering.
>
> To further deepen our understanding of the depth-layering design, we followed the reviewer’s advice and conducted an ablation study using **Grounded-SAM**. Specifically, we semantically segmented each scene into two components, robot and background, with the text prompt _"Robot Arms"_. After segmentation, we applied the standard H$^3$DP pipeline. The results are shown below:
>
> **Table 1: Comparison of depth-based layering with segmentation-based layering.**
> |  Algorithm  | Soccer | Stick Pull | Pick Out of Hole | Fill | Excavate | Tool Adjust |Average|
> | :--- | :---: | :---: | :---: | :---: | :---: | :---: | :---: |
> | Ours  | 85 | 83 |40|98|38|45|64.8|
> | w/ GMM  | 45 | 67 |32|75|27|37|47.2|
> | w/ Grounded SAM  | 63 | 65 |35|79|30|42|52.3|
> | no layering  | 59 | 72 |34|78|27|32|50.3|
>
> **Table 2: Inference speed comparison.**
> | Algorithms   | Inference Speed (FPS)
> | :--- | :---: |
> | Ours | 12.0 |
> | w/ Grounded SAM | 2.6 |
>
> While policies using Grounded-SAM or GMM learn visual representations through semantics-driven layering, our method relies on a fundamentally different principle. The depth-based layering adopted in H³DP does not assume semantic segmentation. Instead, **it leverages the intuition that grouping pixels within the same depth interval provides more coherent information, making it easier for the policy to learn robust visual representations. This operation is therefore distinct from semantic partitioning.** Our experiments support this argument: when applied appropriately, purely depth-based layering guides visuomotor policy learning even better. We have added the experimental results in Table 8 in our revised paper. Moreover, inference-time comparison shows that Grounded-SAM introduces significant latency, greatly limiting its practicality for real-world deployment.

---

> ### Author Response · Authors · 2025-11-24
> **Reply to Reviewer ija8 (2/3)**
>
> A2. We appreciate the reviewer’s suggestion to explore hierarchical action decomposition over longer horizons. At present, H$^3$DP applies hierarchical modeling only to the generation of low-level actions rather than high-level action chunks. Nevertheless, we would like to respectfully emphasize that our carefully designed hierarchy still outperforms baselines that explicitly predict high-level action chunks. To substantiate this claim, we conducted additional experiments comparing H$^3$DP with Dense Policy [1], a representative baseline that first predicts coarse action chunks and then progressively refines them to fine-grained actions.
>
> **Table 3: Comparison with Dense Policy.**
> |  Algorithm   | Door | Pen| Laptop | Bin Picking | Box Close | Hammer | Peg Insert Side | Disassemble | Shelf Place |Average|
> | :--- | :---: | :---: | :---: | :---: | :---: | :---: | :---: | :---: | :---: | :---: |
> | H$^3$DP  | 79 | 83 |81|100|98|100|98|96|100|92.8|
> | 3D Dense Policy  | 72 | 61 |85|47|69|100|82|98|77|76.7|
> | 2D Dense Policy  | 59 | 65 |28|25|51|86|60|71|59|56.0|
>
> As shown above, H$^3$DP consistently outperforms multiple variants of Dense Policy across diverse simulation environments. These results suggest that an appropriately designed hierarchical action-generation paradigm **can achieve superior performance even when it operates solely by predicting and iteratively refining low-level actions**, without explicitly modeling high-level action chunks. We have added the results in Table 10 in the main text of our revised paper.
>
> We would also like to note that the reviewer’s insightful suggestion highlights a promising direction for extending our method. We sincerely appreciate this valuable feedback, and we plan to further investigate how H$^3$DP can be integrated with high-level task decomposition or subtask sequencing in future work.
>
> [1] Yue Su et al., "Dense Policy: Bidirectional Autoregressive Learning of Actions". 2024. https://arxiv.org/abs/2503.13217

---

> ### Author Response · Authors · 2025-11-24
> **Reply to Reviewer ija8 (3/3)**
>
> _Q3. Minor writing issues: Some abbreviations (e.g., MW, MS, RT in Tables 3 and 4) are not defined when first introduced. Clarifying these at the start of the experimental section would improve readability and presentation quality._
>
> A3. Thank you for the suggestion. MW, MS, and RT refer to MetaWorld, ManiSkill, and RoboTwin respectively. We have clarified the meanings of these abbreviations at the beginning of the experimental section to enhance readability.

---

> > ### Comment · Reviewer_ija8 · 2025-11-28
> > **Reply to Authors**
> >
> > Thanks for providing the additional experimental results and analysis, which strengthen the paper. I am also glad to see the authors acknowledge the potential research direction of incorporating high-level actions into the current pipeline. With these clarifications and discussions, most of my concerns have been addressed, and I now support acceptance of the paper.

---

### Official Review · Reviewer_wXbT · 2025-11-10

**Soundness:** 3
**Presentation:** 3
**Contribution:** 3
**Rating:** 6
**Confidence:** 3

**Summary:**

**Motivation**

Previous methods have focused on refining inputs, perception, and action separately. In contrast, humans process information hierarchically at all stages.

**Proposal**

The authors present H3DP – Triply‑Hierarchical Diffusion Policy. An agent with hierarchical processing at three stages of action prediction: inputs, representation and the action prediction.

The agent uses RGB-D inputs, where the images are segmented into multiple layers using the depth maps.

For representation, multiple scaled input images are used to provide information from all levels of granularity.

For action, the diffusion is conditioned progressively from coarse to fine features.

**Results**

The agent is evaluated in sim benchmarks (44 tasks in 5 sims) and real world tasks (4 bimanual manipulation).

The agent performs well in the simulation tasks (75.6%), outperforming the next performing baseline (DP3) by 27.5% (relative) in sim tasks.

The agent also performs well in real world tasks (57.75% overall, +72.4% relative to next baseline).

Additionally, the method performs better than baselines with 20% data.

**Strengths:**

- Good integration of multiple complex architectures: multi-layered depth separated visual inputs, multi-scale image features, and scale-specific feature conditioned diffusion policy.
- Method works in the real world under cluttered conditions (move the bottle in the fridge, pour water and a scoop of powder, sweep trash into the dustbin using a broom and dustpan, and move the bottle to the designated spot).
- Extensive evaluation and ablation study.

**Weaknesses:**

- The evaluation uses only 3 seeds per task. It should ideally be at least 5.
- In some environments, the top 5 success rates are used, and only the top success rate in some (with only 20 episodes in each). This makes it difficult to judge the variance in performance. The number of episodes should be increased (at least 50), and maybe a plot with evaluation scores averaged across all seeds against training step should be reported for a few tasks.
- While the method is tested in general/standard settings, if possible, validation on tasks where the contributions must be required to complete would be great. For example, testing on an occlusion-heavy task, where the depth-wise layering is needed for success.
- The paper does not explicitly mention failure modes or scalability issues. Given the number of hyperparameters (different $\beta$s, learning‑rate schedules, codebook sizes per simulator), a brief discussion on sensitivity and potential deployment challenges would be valuable.

**Questions:**

- How susceptible is the performance to the hyperparameter choice?

---

> ### Author Response · Authors · 2025-11-24
> **Reply to Reviewer wXbT (1/5)**
>
> We thank the reviewer for the positive feedback and constructive suggestions. We address the concerns from the following aspects:
>
> _Q1. The evaluation uses only 3 seeds per task. It should ideally be at least 5._
>
> A1. We agree that more seeds would provide a better estimate of performance variance. We provide additional evaluations with **5** seeds for all tasks in MetaWorld Medium, MetaWorld Hard, MetaWorld Hard++, Adroit, and Dexart. The mean success rates and standard deviations are shown as follows:
>
> **Table 1: Success rates on MetaWorld (Medium) tasks.**
> | Method \ Task | Basketball | Bin Picking | Box Close | Coffee Pull | Coffee Push | Hammer | Peg Insert Side | Push Wall | Soccer | Sweep | Sweep Into | Average |
> | :--- | :---: | :---: | :---: | :---: | :---: | :---: | :---: | :---: | :---: | :---: | :---: | :---: |
> | **Ours** | 100 $\pm$ 0 | 99 $\pm$ 1 | 97 $\pm$ 2 | 100 $\pm$ 0 | 100 $\pm$ 0 | 100 $\pm$ 0 | 97 $\pm$ 1 | 100 $\pm$ 0 | 79 $\pm$ 8 | 100 $\pm$ 0 | 100 $\pm$ 0 | 98 $\pm$ 1 |
> | DP | 100 $\pm$ 0 | 95 $\pm$ 1 | 82 $\pm$ 1 | 81 $\pm$ 1 | 84 $\pm$ 0 | 64 $\pm$ 2 | 61 $\pm$ 1 | 75 $\pm$ 1 | 42 $\pm$ 6 | 95 $\pm$ 1 | 73 $\pm$ 1 | 78 $\pm$ 1 |
> | DP (w/ depth) | 99 $\pm$ 0 | 98 $\pm$ 1 | 77 $\pm$ 2 | 79 $\pm$ 3 | 78 $\pm$ 2 | 63 $\pm$ 3 | 52 $\pm$ 3 | 68 $\pm$ 3 | 34 $\pm$ 4 | 98 $\pm$ 1 | 99 $\pm$ 0 | 77 $\pm$ 2 |
> | DP3 | 100 $\pm$ 0 | 100 $\pm$ 0 | 82 $\pm$ 8 | 100 $\pm$ 0 | 100 $\pm$ 0 | 96 $\pm$ 2 | 90 $\pm$ 6 | 96 $\pm$ 3 | 52 $\pm$ 14 | 100 $\pm$ 0 | 62 $\pm$ 4 | 89 $\pm$ 3 |
>
> **Table 2: Success rates on MetaWorld (Hard) tasks.**
> | Method \ Task | Assembly | Hand Insert | Pick Out of Hole | Pick Place | Push | Average |
> | :--- | :---: | :---: | :---: | :---: | :---: | :---: |
> | **Ours** | 100 $\pm$ 0 | 100 $\pm$ 0 | 37 $\pm$ 4 | 97 $\pm$ 2 | 100 $\pm$ 0 | 87 $\pm$ 1 |
> | DP | 99 $\pm$ 0 | 73 $\pm$ 5 | 12 $\pm$ 2 | 0 $\pm$ 0 | 77 $\pm$ 7 | 52 $\pm$ 3 |
> | DP (w/ depth) | 100 $\pm$ 0 | 74 $\pm$ 3 | 33 $\pm$ 2 | 0 $\pm$ 0 | 79 $\pm$ 1 | 58 $\pm$ 1 |
> | DP3 | 99 $\pm$ 1 | 34 $\pm$ 15 | 31 $\pm$ 6 | 0 $\pm$ 0 | 96 $\pm$ 2 | 52 $\pm$ 5 |
>
> **Table 3: Success rates on MetaWorld (Hard++) tasks.**
> | Method \ Task | Shelf Place | Disassemble | Stick Pull | Stick Push | Pick Place Wall | Average |
> | :--- | :---: | :---: | :---: | :---: | :---: | :---: |
> | **Ours** | 100 $\pm$ 0 | 95 $\pm$ 1 | 80 $\pm$ 4 | 100 $\pm$ 0 | 100 $\pm$ 0 | 95 $\pm$ 1 |
> | DP | 20 $\pm$ 2 | 80 $\pm$ 2 | 63 $\pm$ 1 | 69 $\pm$ 2 | 53 $\pm$ 1 | 57 $\pm$ 2 |
> | DP (w/ depth) | 30 $\pm$ 1 | 77 $\pm$ 3 | 71 $\pm$ 2 | 99 $\pm$ 1 | 80 $\pm$ 1 | 72 $\pm$ 2 |
> | DP3 | 85 $\pm$ 2 | 97 $\pm$ 2 | 59 $\pm$ 2 | 99 $\pm$ 0 | 96 $\pm$ 2 | 88 $\pm$ 2 |
>
> **Table 4: Success rates on Adroit and Dexart tasks.**
> | Method \ Task | Laptop | Faucet | Toilet | Bucket | Hammer | Door | Pen | Average |
> | :--- | :---: | :---: | :---: | :---: | :---: | :---: | :---: | :---: |
> | **Ours** | 83 $\pm$ 4 | 32 $\pm$ 3 | 73 $\pm$ 4 | 27 $\pm$ 2 | 100 $\pm$ 0 | 83 $\pm$ 1 | 78 $\pm$ 3 | 68 $\pm$ 2 |
> | DP | 67 $\pm$ 4 | 21 $\pm$ 4 | 56 $\pm$ 5 | 26 $\pm$ 3 | 94 $\pm$ 2 | 67 $\pm$ 2 | 71 $\pm$ 2 | 58 $\pm$ 3 |
> | DP (w/ depth) | 61 $\pm$ 5 | 19 $\pm$ 2 | 59 $\pm$ 6 | 26 $\pm$ 6 | 99 $\pm$ 1 | 65 $\pm$ 3 | 64 $\pm$ 5 | 56 $\pm$ 4 |
> | DP3 | 82 $\pm$ 4 | 32 $\pm$ 1 | 76 $\pm$ 3 | 28 $\pm$ 2 | 100 $\pm$ 0 | 72 $\pm$ 3 | 80 $\pm$ 4 | 67 $\pm$ 3 |
>
> Some results are slightly different from those in the original submission due to the increased number of seeds. However, the mean success rates remain consistent with our original findings, and H$^3$DP continues to outperform the baselines significantly. We have included the results in Table 35 in revised version of our paper.
>
> Due to the restricted time of discussion period, we are unable to rerun all experiments for 5 seeds for all tasks. We will run all remaining experiments and include these updated results in the final version of the paper.

---

> ### Author Response · Authors · 2025-11-24
> **Reply to Reviewer wXbT (2/5)**
>
> _Q2. The number of episodes should be increased (at least 50), and maybe a plot with evaluation scores averaged across all seeds against training step should be reported for a few tasks._
>
> A2. We appreciate the suggestion to increase the number of evaluation episodes for better variance estimation. We have conducted additional evaluations with 50 episodes for selected tasks across different simulators. A summary of the results is provided below:
>
> **Table 5: Success rates with different number of evaluation episodes.**
> | Method \ Task | Box Close | Pick Place | Disassemble | Basketball | Door | Laptop | Average |
> | :--- | :---: | :---: | :---: | :---: | :---: | :---: | :---: |
> | **Ours (20 eps)** | 98 | 98 | 98 | 100 | 79 | 81 | 92 |
> | **Ours (50 eps)** | 95 | 96 | 93 | 100 | 81 | 82 | 91 |
>
> Besides, we also provide the plot with success rate averaged across all seeds against training step in Figure 9.

---

> ### Author Response · Authors · 2025-11-24
> **Reply to Reviewer wXbT (3/5)**
>
> _Q3. While the method is tested in general/standard settings, if possible, validation on tasks where the contributions must be required to complete would be great. For example, testing on an occlusion-heavy task, where the depth-wise layering is needed for success._
>
> A3. We strongly agree that testing on occlusion-heavy tasks would better demonstrate the necessity of our hierarchical input processing. To fully validate the effectiveness of H$^3$DP, we conduct additional experiments on two special tasks: _occlusion-heavy cluttered_ task (tool adjust), where several objects are stacked together, to show the advantage of depth-aware layering in handling complex spatial arrangements; _dimmed lighting_ task (place bottle), where the ambient light is significantly reduced during inference, to demonstrate the performance of H$^3$DP when depth information is essential.
>
> **Table 6: Success rates under varied lighting conditions.**
> | Dimmed Lighting (%) | 0 | 10 | 20 | 30 | 40 | 50 |
> | :---: | :---: | :---: | :---: | :---: | :---: | :---: |
> | Success Rate (%) | 52 | 55 | 50 | 40 | 45 | 40 |
>
> **Table 7: Success rates of different methods occlusion-heavy cluttered (tool adjust).**
> | Method | DP | DP (w/ depth) | H$^3$DP |
> | :---: | :---: | :---: | :---: |
> | Success Rate (%) | 0 | 32 | 45 |
>
> In the occlusion-heavy cluttered task, H$^3$DP outperforms both DP and DP (w/ depth) by a significant margin, highlighting its ability to effectively utilize depth information in complex scenes. In the dimmed lighting task, H$^3$DP maintains robust performance even as lighting conditions deteriorate, demonstrating the resilience of its hierarchical visual representation. We have included the results as Table 31 and Table 32 in revised version of our paper.

---

> ### Author Response · Authors · 2025-11-24
> **Reply to Reviewer wXbT (4/5)**
>
> _Q4. The paper does not explicitly mention failure modes or scalability issues. Given the number of hyperparameters (different $\beta_s$, learning‑rate schedules, codebook sizes per simulator), a brief discussion on sensitivity and potential deployment challenges would be valuable._
>
> A4. We sincerely thank the reviewer for pointing out the need to discuss scalability. We discuss two main aspects regarding scalability: scale of the model size and scale of expert demonstrations, as follows:
> - Model size: we conduct experiments with different model sizes (small, base) by adjusting the number of layers and hidden dimensions in the model. We provide a comparison of success rates for different model sizes, which has been added as Table 26 in the paper.
>
> **Table 8: Success rates with different model sizes.**
> | Method \ Task | Soccer | Pick Out of Hole | Stick Pull | Laptop | Door | Pen | Average |
> | :--- | :---: | :---: | :---: | :---: | :---: | :---: | :---: |
> | **Small** | 73 | 31 | 72 | 69 | 72 | 78 | 66 |
> | **Base** | **85** | **40** | **83** | **81** | **79** | **83** | **75** |
>
> - Expert demonstrations: we evaluate the performance of H$^3$DP with varying amounts of expert demonstrations in both simulation and real-world settings. We have added the results as Table 27 in the paper.
>
> **Table 9: Success rates with different amounts of expert demonstrations.**
> | Method \ Task | Soccer | Pick Out of Hole | Stick Pull | Laptop | Door | Pen | Average |
> | :--- | :---: | :---: | :---: | :---: | :---: | :---: | :---: |
> | **20% Demos** | 43 | 23 | 64 | 59 | 60 | 48 | 50 |
> | **100% Demos** | 85 | 40 | 83 | 81 | 79 | 83 | 75 |
> | **200% Demos** | **93** | **57** | **88** | **86** | **84** | **86** | **82** |
>
> | Method \ Task | Clean Fridge | Pour Juice | Sweep Trash | Place Bottle | Average |
> | :--- | :---: | :---: | :---: | :---: | :---: |
> | DP | 13 | 24 | 50 | 15 | 23 |
> | DP3 | 12 | 41 | 48 | 33 | 34 |
> | **20% Demos** | 37 | 44 | 58 | 33 | 43 |
> | **100% Demos** | **51** | **65** | **63** | **52** | **58** |
>
> As shown in the tables above, both increasing model size and the amount of expert demonstrations lead to improved performance. Besides, with only 20% of expert demonstrations, H$^3$DP still achieves comparable or better performance than baselines with full demonstrations, showcasing its data efficiency. We have included these discussions and results in Appendix E.10 of the revised version of the paper.

---

> ### Author Response · Authors · 2025-11-24
> **Reply to Reviewer wXbT (5/5)**
>
> _Q5. How susceptible is the performance to the hyperparameter choice?_
>
> A5. Thank you for the question. We would like to clarify that in H$^3$DP, only the ManiSkill experiments use a different learning-rate scheduler (One-Cycle LR) and different AdamW $\beta_s$ values ($\beta_s=(0.9, 0.95)$), while all other simulation benchmarks adopt the cosine scheduler and $\beta_s=(0.95, 0.999)$. **This choice is made solely to remain consistent with previous work rather than as a deliberate or carefully tuned design choice.** The same guideline, which is to align with the original environment settings, applies to the other simulation tasks as well.
>
> To address the reviewer’s concern, we conducted additional ablation studies on MetaWorld to assess the influence of the learning-rate scheduler and $\beta_s$ settings. Specifically, we evaluated multiple schedulers (Cosine, Constant, and One-Cycle LR). The results are shown below. Moreover, since we previously performed extensive experiments on MetaWorld to determine the codebook size, we directly reuse those prior results, which cover substantially more tasks than our other ablations.
>
> The results indicate that both the learning-rate scheduler and $\beta_s$ values have only a minor impact on overall performance. Regarding the codebook size, although 64 yields the best results, the performance of 32 and 128 remains comparable, suggesting that this hyperparameter is not highly sensitive. **These findings demonstrate that H$^3$DP is robust to variations in hyperparameters.**
>
> **Table 10: Hyperparameter sensitivity results on MetaWorld tasks.**
> | $\beta_s$   | Assembly | Pick Place | Hand insert | Pick Out of Hole | Push | Average |
> | :--- | :---: | :---: | :---: | :---: | :---: | :---: |
> |  (0.95, 0.999) (ours)  |100| 99|100 |40 |100 |88|
> |  (0.9, 0.95)  | 100 | 90|100 |33 |95 |84|
>
> | Learning Rate Scheduler   | Assembly | Pick Place|Hand insert |Pick Out of Hole |Push|Average|
> | :--- | :---: | :---: | :---: | :---: | :---: | :---: |
> |  Cosine (ours)  |100| 99|100 |40 |100 |88|
> |  Constant ($lr=10^{-4}$)  | 100 | 100|96 |33 |98 |85|
> |  Constant ($lr=10^{-5}$)  | 100 | 97|99 |30 |88 |83|
> |  One Cycle LR  | 100 | 97|94 |30 |90 |82|
>
> | Codebook Size   | Assembly | Pick Place|Hand insert |Pick Out of Hole |Push |Shelf Place |Disassemble|Stick Pull|Soccer|Sweep into|Bin Picking|Pick Place Wall|Average|
> | :--- | :---: | :---: | :---: | :---: | :---: | :---: | :---: | :---: | :---: | :---: | :---: | :---: | :---: |
> | 64 (ours)  | 100 | 99 |100|40|100|100|96|83|85|100|100|100|92|
> | 32  | 100 | 92 |90|38|100|100|97|78|83|100|93|99|89|
> | 128  | 100 | 88 |93|40|100|100|98|75|59|100|100|100|88|

---

### Official Review · Reviewer_CE83 · 2025-11-11

**Soundness:** 3
**Presentation:** 3
**Contribution:** 3
**Rating:** 8
**Confidence:** 3

**Summary:**

This paper proposes H3DP (Triply-Hierarchical Diffusion Policy), a hierarchical generative framework for visuomotor policy learning in robotic manipulation. The method integrates three levels of hierarchy: depth-aware input layering that partitions RGB-D observations, multi-scale visual representations that preserve global-to-local semantics, and a coarse-to-fine action generation diffusion process. Extensive experiments are conducted across 44 simulated tasks (5 benchmarks) and 4 challenging real-world bimanual manipulation tasks, demonstrating significant performance gains over major baselines and ablation variants.

**Strengths:**

1.  **Ablation and analysis depth:**  The authors ablate each hierarchical design aspect (see Tables 13–15), show model behaviors under low-data and noisy depth regimes (Tables 7, 22), and compare to closely-related algorithmic variants (DP3, CARP). The connection between model robustness and architectural choices (quantization, depth-layering) is examined through concrete experiments and Figure 7.
2.  **Efficiency:**  H3DP achieves results rivaling or surpassing point-cloud and multi-view methods but with only single-view RGB-D as input, reducing computational and engineering demands. Model size and inference speed are competitive (Tables 20–21).
3. **Empirical evaluation:**  The paper covers a broad suite of tasks: 44 simulated and 4 real-world challenges, while using established benchmarks (MetaWorld, ManiSkill, Adroit, etc.) and challenging bimanual/long-horizon settings. The use of both instance and spatial generalization metrics is appreciated. Table 1 and Figure 1 provide clear, accessible summaries of the breadth and magnitude of gains.

**Weaknesses:**

1.  **Incomplete baselines:** I think the main weakness lies in the incomplete set of baselines used for comparison. The model is primarily evaluated against DDPM-based approaches, making it unclear how well it performs relative to flow-based [1, 2, 3] or equivariance-based designs [4, 5, 6].

[1] Zhang et al., "FlowPolicy: Enabling Fast and Robust 3D Flow-based Policy via Consistency Flow Matching for Robot Manipulation". 2024. https://arxiv.org/abs/2412.04987

[2] Chisari et al., "Learning Robotic Manipulation Policies from Point Clouds with Conditional Flow Matching". 2024. https://arxiv.org/abs/2409.07343

[3] Ding et al., "Fast and Robust Visuomotor Riemannian Flow Matching Policy". 2025. https://doi.org/10.1109/TRO.2025.3601293

[4] Wang et al., "Equivariant Diffusion Policy". 2024. https://openreview.net/forum?id=wD2kUVLT1g

[5] Yang et al., "EquiBot: SIM(3)-Equivariant Diffusion Policy for Generalizable and Data Efficient Learning". 2024. https://arxiv.org/abs/2407.01479

[6] Tie et al., "ET-SEED: Efficient Trajectory-Level SE(3) Equivariant Diffusion Policy". 2024. https://arxiv.org/abs/2411.03990

**Questions:**

1. See weaknesses.
2. How were the depth maps obtained? It would be interesting to evaluate how well the method performs when the depth maps are generated by a neural model from RGB images.

---

> ### Author Response · Authors · 2025-11-24
> **Reply to Reviewer CE83 (1/2)**
>
> We thank the reviewer for the positive feedback and constructive suggestions. We address the concerns from the following aspects:
>
> _Q1. Incomplete baselines: I think the main weakness lies in the incomplete set of baselines used for comparison. The model is primarily evaluated against DDPM-based approaches, making it unclear how well it performs relative to flow-based [1, 2, 3] or equivariance-based designs [4, 5, 6]._
>
> A1. Thank you for the valuable suggestion. We agree that including more baselines would strengthen the evaluation of our method. Following the reviewer’s advice, we conducted additional experiments with more baselines. Specifically, we incorporated three new methods, including **FlowPolicy** (flow-based), **ET-SEED** (equivariance-based), and **3D-Diffuser-Actor** (a 3D-aware representation learning policy), and evaluated them on the MetaWorld-Hard tasks. Since 3D-Diffuser-Actor demands task prompts to guide the policy, we provided precise and unambiguous textual prompts as its language inputs, such as _"push the red object to the green target"_ for MetaWorld-Push and _"place the cube on the second floor of the shelf"_ for MetaWorld-ShelfPlace.
>
> As shown in the table below, H$^3$DP consistently outperforms all three additional baselines across the evaluated benchmarks, demonstrating its strong capability to learn effective visual representations and handle challenging simulation tasks. We have added the experiment results in Table 9 in our paper. Moreover, we are grateful to the baselines mentioned by the reviewer and have referred to all suggested works in the revised version of the paper.
>
> **Table 1: Success rates on MetaWorld-Hard tasks with additional baselines.**
> | Algorithms   | Assembly | Pick Place| Shelf Place|Hand insert |Pick Out of Hole |Push|Average|
> | :--- | :---: | :---: | :---: | :---: | :---: | :---: | :---: |
> |  H$^3$DP  |100| 99|100|100 |40 |100 |89.8|
> |  FlowPolicy  |100| 12|83|42 |29 |100 |61.0|
> |  ET-SEED  | 100 | 23    |79|89 |37 |96 |70.7|
> |  3D Diffuser Actor  | 100 | 0|65| 33 |26 |100 |54.0|
> |  DP3  | 100 | 0|86|37 |30 |96 |58.2|

---

> ### Author Response · Authors · 2025-11-24
> **Reply to Reviewer CE83 (2/2)**
>
> _Q2.1. How were the depth maps obtained?_
>
> A2.1. We appreciate the reviewer’s interest in the depth map acquisition process. In our experiments, we utilized the built-in functionalities of the simulation environments to obtain depth maps. In real-world applications, depth maps are collected by ZED cameras, as mentioned in Sec. 4.2.1.
>
> _Q2.2. It would be interesting to evaluate how well the method performs when the depth maps are generated by a neural model from RGB images._
>
> A2.2. We sincerely thank the reviewer's insightful question. We adopt **Depth Anything-v3**, a latest-released model to generate depth maps from RGB images. Specifically, we choose the _"BASE"_ and _"SMALL"_ versions of Depth Anything 3 to balance the depth quality as well as the inference speed, since we have to obtain depth maps instantly during inference phase.
>
> We conduct experiments in MetaWorld tasks to show the performance of H$^3$DP with Depth Anything. We also make comparison against Diffusion Policy with depth maps, whose results are shown below.
>
> **Table 2: Success rates on MetaWorld tasks with depth maps generated by Depth Anything-v3.**
> | Algorithms   | Assembly | Shelf Place|Hand insert |Pick Out of Hole |Push|Average|
> | :--- | :---: | :---: | :---: | :---: | :---: | :---: |
> |  Ours  |100| 100|100 |40 |100 |88.0|
> |  w/ Depth Anything 3 (SMALL)  | 100 | 88|88 |35 |55 |73.2|
> |  w/ Depth Anything 3 (BASE)  | 100 | 93|96 |32 |68 |77.8|
> |  DP w/ Depth  | 100 | 29|75 |32 |79 |63.0|
>
> As shown above, Depth-Anything-3 has a positive impact on generating proper depth maps, since H$^3$DP with different versions of Depth-Anything-3 also show fair performance. Nevertheless, they still lag behind original H$^3$DP. A possible reason is that depth maps generated by neural networks often contain inconsistencies at the pixel level, which can degrade visual representation learning. In particular, models such as Depth Anything do not take temporal information as input. Therefore, the predicted depth of the same pixel can vary significantly across consecutive frames. Although such variations may be subtle to human eyes, they can have a substantial negative effect on model training. This observation aligns with Yuan et al. [1], who reported similar issues in Appendix. L. We include these results and analyses in the revised version of the paper (See Appendix E.4 and Table 20)
>
> [1] Yuan et al., "HERMES: Human-to-Robot Embodied Learning from Multi-SouRce Motion Data for MobilE DexterouS Manipulation". 2025. https://arxiv.org/pdf/2508.20085

---

> > ### Comment · Reviewer_CE83 · 2025-11-26
> >
> > Thank you for your detailed response and the additional experiments. You have addressed all of my concerns. I now consider the submission to be strong and believe it should be accepted. I will keep my overall score unchanged, but I am increasing my confidence score.

---

### Author Response · Authors · 2025-11-24
**General Response**

We express our gratitude to the reviewers for their detailed comments and constructive suggestions, as well as for their recognition of the merits of our approach. All the reviewers appreciate the comprehensive experiments, the clarity of the presentation, and the thoroughness of the analysis.

**We have highlighted the changes in blue in the revised version of our paper.** Here, we provide an overview of our changes:

- **(i) Inclusion of additional baselines:** We have further compared H$^3$DP against a flow-based policy (FlowPolicy), an equivariance-based method (ET-SEED), another 3D-aware representation learning policy (3D-Diffuser-Actor), and an action hierarchical decomposition method (Dense Policy). Across all evaluated benchmarks, H$^3$DP consistently outperforms these four additional baselines, demonstrating its strong capability and generalizability to handle diverse tasks. The corresponding results have been added to Section 4.4.
- **(ii) Increased number of random seeds:** We have increased the number of random seeds to 5 for each experiment and report the standard deviation for each task.
- **(iii) Comparison with semantic segmentation approaches:**  We have  included a new comparison with a segmentation-based layering approach built upon Grounded SAM. The experiments show that our depth-based layering method achieves better performance than explicitly performing semantic segmentation for layering. The analysis and results are presented in Section 4.3.2.
- **(iv) Additional tasks with varying lighting conditions:** We further evaluate H$^3$DP on tasks involving changes in lighting conditions. H$^3$DP remains robust under these challenging visual perturbations.
- **(v) Additional experiments with neural depth estimation models:** We further evaluate H$^3$DP using depth maps predicted by Depth Anything-v3. Under this setting, H$^3$DP with estimated depth still achieves better performance than the baselines.
- **(vi) Ablation study on hyperparameters:** We have conducted experiments to test the sensitivity of hyperparameters in H$^3$DP. Specifically, we vary the value of $\beta_s$, learning-rate schedules and codebook sizes and record the performance. The results show the resilience H$^3$DP has to the shift of hyperparameters.

In addition, we conducted several further experiments, including scalability analysis, ablation on codebook sharing and evaluations with an increased number of episodes, etc., to address the reviewers' concerns.

---

### Meta-Review · Area_Chair_XLPk · 2026-01-08

**Summary:**

This paper proposes H3DP (Triply-Hierarchical Diffusion Policy), a hierarchical generative framework for visuomotor policy learning in robotic manipulation. The method integrates three levels of hierarchy. Extensive experiments are conducted across 44 simulated tasks (5 benchmarks) and 4 challenging real-world bimanual manipulation tasks, demonstrating significant performance gains over major baselines and ablation variants.

**Reviewer Concerns:**

- Incomplete baselines
- Insufficient experiments / clarity on experimental setups / failure modes
- Comparison on different Methodology. Technical details (e.g. hierarchical actions)
- Performance clarity and fairness / more detailed evaluation / clarity on evaluation metrics
- Lack of baselines (again)
- Technical details

**Reviewer Scores:**

This paper receives 6 reviews and three of them have responded in reply with positive feedback. All six reviews reach concensus that the paper is polished well.

AC read the paper, experiments, review, rebuttal. AC believes the paper is in good shape.

---

### Decision · Program_Chairs · 2026-01-26

Accept (Poster)